# PHD1 controls muscle mTORC1 in a hydroxylation-independent manner by stabilizing leucyl tRNA synthetase

Gommaar D'Hulst[1,13], Inés Soro-Arnaiz[1,13], Evi Masschelein[1], Koen Veys [2,3], Gillian Fitzgerald[1], Benoit Smeuninx[4,5], Sunghoon Kim [6], Louise Deldicque[7], Bert Blaauw[8], Peter Carmeliet [2,3], Leigh Breen[4,5], Peppi Koivunen[9], Shi-Min Zhao[10,11,12] & Katrien De Bock [1]*

mTORC1 is an important regulator of muscle mass but how it is modulated by oxygen and nutrients is not completely understood. We show that loss of the prolyl hydroxylase domain isoform 1 oxygen sensor in mice (PHD1[KO]) reduces muscle mass. PHD1[KO] muscles show impaired mTORC1 activation in response to leucine whereas mTORC1 activation by growth factors or eccentric contractions was preserved. The ability of PHD1 to promote mTORC1 activity is independent of its hydroxylation activity but is caused by decreased protein content of the leucyl tRNA synthetase (LRS) leucine sensor. Mechanistically, PHD1 interacts with and stabilizes LRS. This interaction is promoted during oxygen and amino acid depletion and protects LRS from degradation. Finally, elderly subjects have lower PHD1 levels and LRS activity in muscle from aged versus young human subjects. In conclusion, PHD1 ensures an optimal mTORC1 response to leucine after episodes of metabolic scarcity.

[1] Department Health Sciences and Technology, Laboratory of Exercise and Health, Swiss Federal Institute of Technology (ETH) Zurich, Zurich, Switzerland. [2] Laboratory of Angiogenesis and Vascular Metabolism, VIB Center for Cancer Biology (CCB), VIB, Leuven, Belgium. [3] Laboratory of Angiogenesis and Vascular Metabolism, Department of Oncology, KU Leuven, Leuven, Belgium. [4] School of Sport, Exercise and Rehabilitation Sciences, University of Birmingham, Edgbaston, Birmingham B15 2TT, UK. [5] MRC Arthritis Research UK Centre for Musculoskeletal Ageing Research, University of Birmingham, Edgbaston, Birmingham B15 2TT, UK. [6] Medicinal Bioconvergence Research Center, College of Pharmacy, Seoul National University, Gwanak-gu, Seoul, South Korea. [7] Institute of Neuroscience, Université catholique de Louvain, Louvain-La-Neuve, Belgium. [8] Department of Biomedical Sciences, Venetian Institute of Molecular Medicine, University of Padova, Padova, Italy. [9] Biocenter Oulu, Faculty of Biochemistry and Molecular Medicine, Oulu Center for Cell-Matrix Research, University of Oulu, Oulu, Finland. [10] Obstetrics and Gynaecology Hospital of Fudan University, State Key Lab of Genetic Engineering, School of Life Sciences and Institutes of Biomedical Sciences, Shanghai, P. R. China. [11] Institute of Biomedical Science, Fudan University, Shanghai, P. R. China. [12] Collaborative Innovation Center for Biotherapy, West China Hospital, Sichuan University, Chengdu, P. R. China. [13]These authors contributed equally: Gommaar D'Hulst, Inés Soro-Arnaiz. *email: katrien-debock@ethz.ch

Skeletal muscle mass is essential to life as it provides mechanical power for movement and at the same time plays a crucial role in whole body metabolism. From the age of 50, skeletal muscle mass is lost at a rate of 1–2% per year[1], resulting in diminished functional strength that correlates with lower overall quality of life and increased mortality[2,3]. Muscle is also critical component of diseases such as chronic obstructive pulmonary disorder, diabetes, cancer, anemia, and sepsis[4–6]. The clinical and financial burden of loss of muscle mass to society is enormous[7], and it is therefore vital to develop strategies to prevent loss of muscle mass, or to maintain and even increase muscle mass. To do so, we need to improve our understanding of the underlying molecular mechanisms that control skeletal muscle mass.

Skeletal muscle mass is defined by a fine balance between protein synthesis and breakdown[8,9], processes which are governed by mechanistic target of rapamycin complex 1 (mTORC1), a master regulator of cellular metabolism[10]. Growth factors, energetic stress, and contractions control mTORC1 directly or via its upstream inhibitor tuberous sclerosis complex 1/2 (TSC1/2)[11,12]. Amino acids regulate mTORC1 via alternative pathways involving Rag GTPases, which recruit mTORC1 to the lysosomal membrane[13,14]. How TSC2-dependent signals regulate mTORC1 in skeletal muscle has been intensely studied[11,15,16], but how fluctuations of amino acids, and in particular the essential amino acid leucine, regulate mTORC1, and thus skeletal muscle mass in vivo is much less well understood.

Over the last years, intense research efforts have led to the discovery of leucine sensors, such as SESTRINs (SESNs) and leucyl-tRNA synthetase (LRS), which transmit intracellular leucine availability toward mTORC1[17,18]. SESNs negatively regulate mTORC1 by binding and inhibiting GATOR2, an upstream activator of mTORC1[19]. Upon leucine stimulation, SESN2 dissociates from GATOR2 resulting in increased mTORC1 activation[17]. LRS, an aminoacyl tRNA synthetase that catalyzes the ligation of leucine to its cognate tRNA[20], also exerts a non-canonical role and activates mTORC1 upon leucine binding by functioning as a GAP toward RagD and by ensuring the leucylation of RagA/B[18,21]. Besides their sensing activities, it is plausible that alterations in protein levels of leucine sensors can impose an additional level of control on mTORC1 activity[22]. For instance, amino acid starvation and stress upregulate SESN2 protein content[23,24], and lead to more SESN2 bound to GATOR2 and thus sustained mTORC1 inhibition. Whether fluctuations in nutrient availability also affect the levels or stability of LRS and whether dynamic changes in LRS content can regulate mTORC1 activity remains to be elucidated.

De novo protein synthesis is a highly energy consuming process[25], while the energy cost of protein breakdown has been estimated to be much lower to even neglectable[26]. Protein synthesis and mTORC1 activity therefore need to be tightly controlled upon metabolic stress such as hypoxia and nutrient deprivation, to ensure cell survival. Dependent on the specific context, hypoxia either strongly inhibits or promotes mTORC1 activity[27–30]. Yet, the cellular machinery that ensures cell survival and orchestrates a fast and efficient cellular recovery upon restoration of oxygen and nutrient supply needs to be maintained. An excellent set of candidates to orchestrate such adaptive responses are prolyl-hydroxylase domain proteins 1–3 (PHD1–3). PHDs belong to a family of enzymes which use oxygen, $Fe^{2+}$, ascorbic acid, and the TCA intermediate α-ketoglutarate (αKG) to hydroxylate proline residues in their target substrate proteins, the best characterized ones being the hypoxia-inducible factors 1–3α (HIF1–3α)[31,32]. During hypoxia, HIF1α inhibits mTORC1 by activating the transcription its downstream target regulated in development and DNA damage response 1 (REDD1) which has

been shown to activate TSC2[15,33]. On the other hand, HIF2α increases mTORC1 activity under low amino acid availability by increasing the expression of the LAT1 amino acid carrier[29]. Besides controlling HIF stability, PHDs can also change the activity and stability of other proteins, either or not in a hydroxylation dependent fashion[34]. But whether PHDs can directly control protein synthesis, is not known.

Here, we show that PHD1 controls the activation of mTORC1 in response to leucine in a hydroxylation-independent manner. PHD1 interacts with and determines the stability of the leucine sensor LRS. This interaction is promoted under conditions of hypoxia and amino acid starvation (when PHD1 activity is inhibited) and protects LRS from degradation. In this way, PHD1 ensures effective activation of mTORC1 in response to leucine. As a consequence, genetic loss of *Phd1* reduces the stability of LRS, impairs leucine mediated activation of mTORC1 and leads to lower muscle mass. The relevance of our data is underscored by the observation that elderly humans have lower PHD1 levels and LRS activity in muscle.

## Results

**_Phd1_-deficient mice have lower muscle mass.** To study the role of PHD1 in mTORC1 activation and muscle mass control in vivo we used PHD1[KO] mice (50% Swiss/50% 129S1)[35]. Both males (Fig. 1a) and females (Supplementary Fig. 1a) showed lower mass of both *m. gastrocnemius* (GAS), *m. tibialis anterior* (TA) and *m. extensor digitorum longus* (EDL). Magnetic resonance imaging (MRI) analysis confirmed that PHD1[KO] mice have lower lean mass when compared with the corresponding controls (Fig. 1b). The reduction in lean mass resulted into lower body weight in males but not in females (Fig. 1d and Supplementary Fig. 1d), where the reduction in lean mass was completely compensated by an increase in fat mass (Supplementary Fig. 1c). This data confirms previous work reporting increased white adipose tissue mass in PHD1[KO] mice[36]. Analysis of fiber area in TA showed that lower muscle weight was accompanied with decreased fiber cross-sectional area (Fig. 1e). Differences in fiber cross-sectional area were not secondary to a shift in muscle fiber type composition (Supplementary Fig. 1e). We also did not find evidence for overt myopathy, indicated by the absence of centrally nucleated fibers (Supplementary Fig. 1f). Absolute force–frequency analysis of ex vivo contracted *soleus* showed reduced force production in PHD1[KO] compared to WT mice (Fig. 1f). Relative force–frequency, which is corrected for muscle surface area, was unaffected (Supplementary Fig. 1h), further confirming that the lower force production in these muscles was caused by lower fiber area and likely not by defective intrinsic mechanical capacities.

Since muscle fiber size is determined by the balance between protein synthesis and protein breakdown, we first monitored the status of the ubiquitin-proteasome and autophagy-lysosome systems, the two main contributors to muscle protein breakdown[37]. RT-qPCR analysis failed to show increased expression of the ubiquitin-proteasome related genes *Atrogin-1*, *Murf1*, *Itch*, *Smart*, *Musa1*, and *FbxO31* (Fig. 1g). mRNA levels of autophagy related genes such as *P62*, *Lc3b*, *GabarapL*, *Bnip3*, and *CathL* were also not affected by loss of *Phd1* (Supplementary Fig. 1g). Accordingly, expression of microtubule-associated protein 1 light chain 3 (LC3-I) and lipidated LC3 (LC3-II) was not different between PHD1[KO] compared to WT muscles which were harvested after 4 h of food withdrawal (Fig. 1h), neither did we find differences in P62 protein content, a marker for autophagy impairment (Fig. 1h). This data indicates that loss of *Phd1* does not substantially promote muscle protein breakdown and prompted us to evaluate whether PHD1 controls muscle protein synthesis.

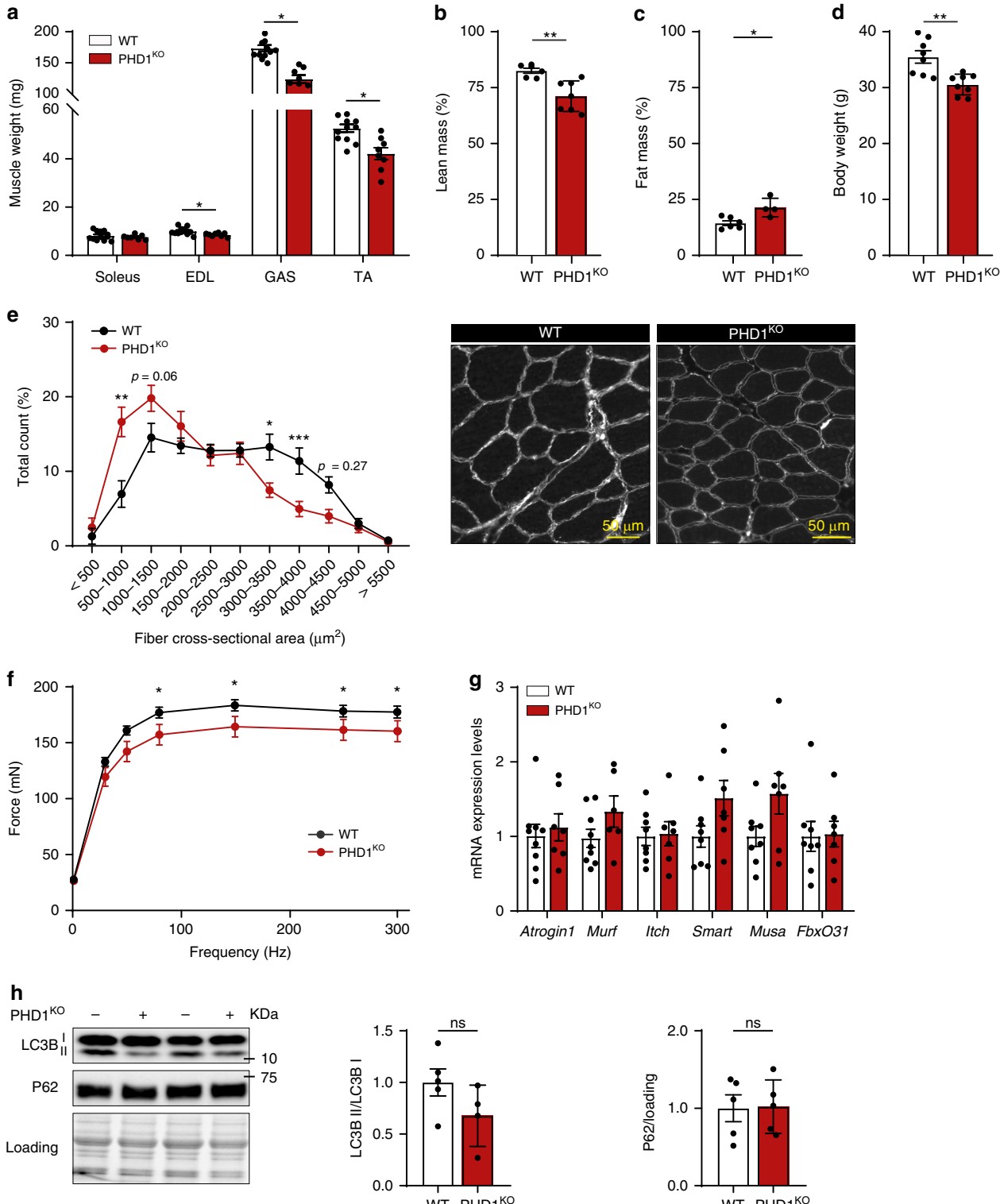

**Fig. 1 *Phd1*-deficient mice have lower muscle mass.** Bar graphs showing muscle weight (**a**), lean mass (**b**), fat mass (**c**), and body weight (**d**) of WT (white bars) and PHD1KO (red bars) male mice. **e** Quantification (left panel) and representative pictures (right panel) of fiber cross sectional area distribution in WT (black line) and PHD1KO (red line) TA muscle. **f** Force–frequency curve in ex vivo stimulated *soleus* from WT (black line) and PHD1KO (red line) male mice. **g** mRNA expression levels of genes involved in ubiquitin-proteasome mediated protein degradation in TA muscle from WT (white bars) and PHD1KO (red bars) female mice. **h** Representative pictures and quantification of western blot analysis of LC3B and P62 protein levels in TA muscle from WT (white bars) and PHD1KO (red bars) female mice. Statistics: two-way ANOVA test, with a Holm–Sidak post hoc test (**e**, **f**) or unpaired *t* test (**a**, **b**, **c**, **d**, **g**, **h**) (*$p < 0.05$; **$p < 0.01$; ***$p < 0.001$; ns not significant). Each dot represents a single mouse (**a**, **b**, **c**, **d**, **h**). Bar graphs and line graphs represent mean ± SEM (error bars). Data is represented as fold change to WT (**g**, **h**). EDL *m. extensor digitorum longus*, GAS *m. gastrocnemius*, TA *m. tibialis anterior*. See also Supplementary Fig. 1. Source data are provided as a Source Data file.

**PHD1 is required for leucine mediated mTORC1 activation.** mTORC1 represents a main regulatory hub in the control of muscle protein synthesis in response to many anabolic signals, such as growth factors, eccentric contractions, and/or amino acids[11,38]. To study whether loss Phd1 affects amino acid mediated activation of mTORC1 in muscle, we administered L-leucine (leucine), the most potent amino acid stimulator of mTORC1 and required for activation of muscle protein synthesis in vivo[39], to PHD1[KO] and WT animals via oral gavage and subsequently analyzed mTORC1 activity. In WT TA muscle, leucine administration activated mTORC1, as judged by the increased phosphorylation states of its substrates S6 kinase 1 (p-S6K1), S6 ribosomal protein (p-RPS6), and the 4E-binding protein 1 (p-4E-BP1) (Fig. 2a and Supplementary Fig. 2a). In contrast, leucine-mediated phosphorylation of these mTORC1 substrates was abrogated in PHD1[KO] muscle (Fig. 2a and Supplementary Fig. 2a). Immunofluorescent quantification of p-RPS6 levels in TA muscle confirmed these observations (Fig. 2b, c). Inhibition of leucine mediated mTORC1 activation upon deletion of Phd1 was observed in both females (Fig. 2a) as well as males (Supplementary Fig. 2a), so both genders were used for subsequent experiments. Moreover, inhibition of mTORC1 activity upon loss of Phd1 is fiber type independent, because we observed impaired mTORC1 activation in soleus, which is predominantly composed of a slower fibers, as well as EDL, which contains more fast glycolytic fibers (Supplementary Fig. 2b, c). To confirm that impaired mTORC1 activation resulted into impaired protein synthesis, we used SunSET analysis and measured puromycin incorporation into muscle protein after leucine injection. This data showed 50% lower protein synthesis in PHD1[KO] muscle when compared to WT (Fig. 2d).

Growth factors such as insulin but also eccentric contractions are also potent regulators of mTORC1 in the muscle[38,40], albeit via different and independent upstream mechanisms[41]. To explore whether PHD1 also controls insulin mediated activation of mTORC1, we injected mice with insulin (0.2 IU/g) and assessed mTORC1 activation. We first confirmed that insulin effectively activated the insulin signaling cascade by assessing the phosphorylation of AKT at Ser[473] (p-AKT) (Supplementary Fig. 2d). In addition, insulin equally increased p-S6K1 in WT and PHD1[KO] animals, indicating that PHD1 controls leucine, but not insulin, mediated activation of mTORC1 (Supplementary Fig. 2d). We subsequently isolated EDL muscle from PHD1[KO] and WT mice and subjected them to an ex vivo eccentric contraction protocol which is known to activate mTORC1 directly[42] or via the inactivation TSC2[11]. Eccentric contractions effectively activated the stress-responsive C-Jun N-terminal kinase (JNK), a key regulator of adaptive remodeling after resistance training[43] in both WT and PHD1[KO] mice (Supplementary Fig. 2e). Moreover, eccentric contractions induced a similar activation of mTORC1 downstream signaling in both WT and PHD1[KO] mice (Supplementary Fig. 2e). Thus, PHD1 controls leucine mediated, but not insulin nor contraction mediated activation of mTORC1 in the muscle.

Growth factors as well as eccentric contractions activate mTORC1 mainly through the inhibition of its inhibitory TSC complex[11,44], the latter being a direct target of HIF-dependent inhibition of protein synthesis via REDD1[15]. Moreover, lack of oxygen availability leads to an AMPK dependent phosphorylation of the TSC complex as well as RAPTOR[45], leading to a general inhibition of protein synthesis. We did not find increased AMPK activation (Fig. 2e) nor phosphorylation of TSC2 at Ser[1387] (Fig. 2f) in PHD1[KO] muscle. We also did not pick up increased expression of the HIF target Redd1 (Fig. 2g). Our data suggest that PHD1 controls mTORC1 via selectively altering its response to leucine. We thus decided to further explore the role of PHD1 in leucine metabolism.

**PHD1 controls muscle mass in a cell-autonomous fashion.** To study whether the blunted leucine mediated mTORC1 activation upon Phd1 deletion was driven by muscle-intrinsic factors, we decided to cross Phd1[fl/fl] mice with human skeletal muscle α-actin (HSA)-Cre-ER[T2] mice[46], which upon tamoxifen treatment results into the generation of muscle specific Phd1 knockout mice (PHD1[mKO]) (Fig. 3a). To evaluate the efficiency of our tamoxifen regimen, we also crossed HSA-Cre-ER[T2] mice with Rosa[mTmG] mice[47], a double fluorescent Cre reporter line that expresses membrane-targeted Tomato (mT) prior to Cre-mediated excision and GFP (mG) after excision, and confirmed efficient recombination which was restricted to skeletal muscles (Supplementary Fig. 3a). Indeed, one week after the last tamoxifen injection, Phd1 mRNA levels in muscle of PHD1[mKO] mice were around 90% lower when compared to littermate controls (Fig. 3b).

We next confirmed that loss of Phd1 in muscle suffices to blunt the activation of mTORC1 downstream targets following leucine but not insulin stimulation (Fig. 3c and Supplementary Fig. 3b). Moreover, leucine-induced protein synthesis assessed via puromycin incorporation was reduced as well (Supplementary Fig. 3c). To further confirm the cell autonomous role of PHD1 in the control of leucine mediated mTORC1 activation, we isolated muscle stem cells from WT and PHD1[KO] mice and differentiated them to myotubes. Myotubes were starved of amino acids and stimulated with leucine. In WT myotubes, leucine dose dependently activated mTORC1, and maximal activation was reached at 5 mM (Fig. 3d). Loss of Phd1 reduced the mTORC1 response at all leucine concentrations tested (Fig. 3d). Off note, we also measured lower mTORC1 activity under full medium conditions (containing 0.8 mM leucine), but consistent with our in vivo data, the response to insulin was preserved (Supplementary Fig. 3d).

**PHD1 controls mTORC1 in a hydroxylation-independent manner.** PHDs can control the stability of proteins by hydroxylating proline residues which targets them for proteasomal degradation[48,49]. To evaluate whether PHD1 controls mTORC1 activity via hydroxylation dependent or independent mechanisms, we transduced primary satellite cells from PHD1[KO] mice with retroviruses to reintroduce either the full length PHD1 (PHD1[WT]) or a catalytically inactive PHD1 mutant (PHD1[MUT])[50]. Introduction of PHD1[WT] as well as PHD1[MUT] in PHD1[KO] myotubes restored leucine-dependent mTORC1 activation (Fig. 3e), showing that PHD1 controls leucine-mediated mTORC1 activation in a hydroxylation-independent manner. Based on these observations, we decided to evaluate how PHD1 controls leucine mediated mTORC1 activation and hypothesized that this could occur through regulating leucine uptake or through interacting with leucine sensing mechanisms[10,13].

**Loss of Phd1 does not impair leucine uptake.** To act on mTORC1 and induce growth, large neutral amino acids such as leucine, enter the cell via coupled amino acid transport[51]. Since leucine stimulated mTORC1 activation was blunted in PHD1[KO] muscle, we investigated whether leucine uptake was impaired in PHD1[KO] mice. First, we evaluated expression levels of main muscle amino acid transporters Snat, Pat1, and Lat1, but these were unaffected in PHD1[KO] mice (Fig. 3f). Second, oral administration of leucine did not alter blood leucine levels (Fig. 3g). And lastly, we measured leucine uptake into the muscle using 1.5 μCi L-[$^{14}$C(U)]-leucine tracer labeling. Uptake of $^{14}$C-leucine by GAS and TA muscle was identical between WT and PHD1[KO] mice (Fig. 3h). This data shows that leucine transport is not impaired in PHD1[KO] mice, and that reduced leucine mediated activation of mTORC1 is likely due to defects in the intracellular leucine-mTORC1 activation cascade.

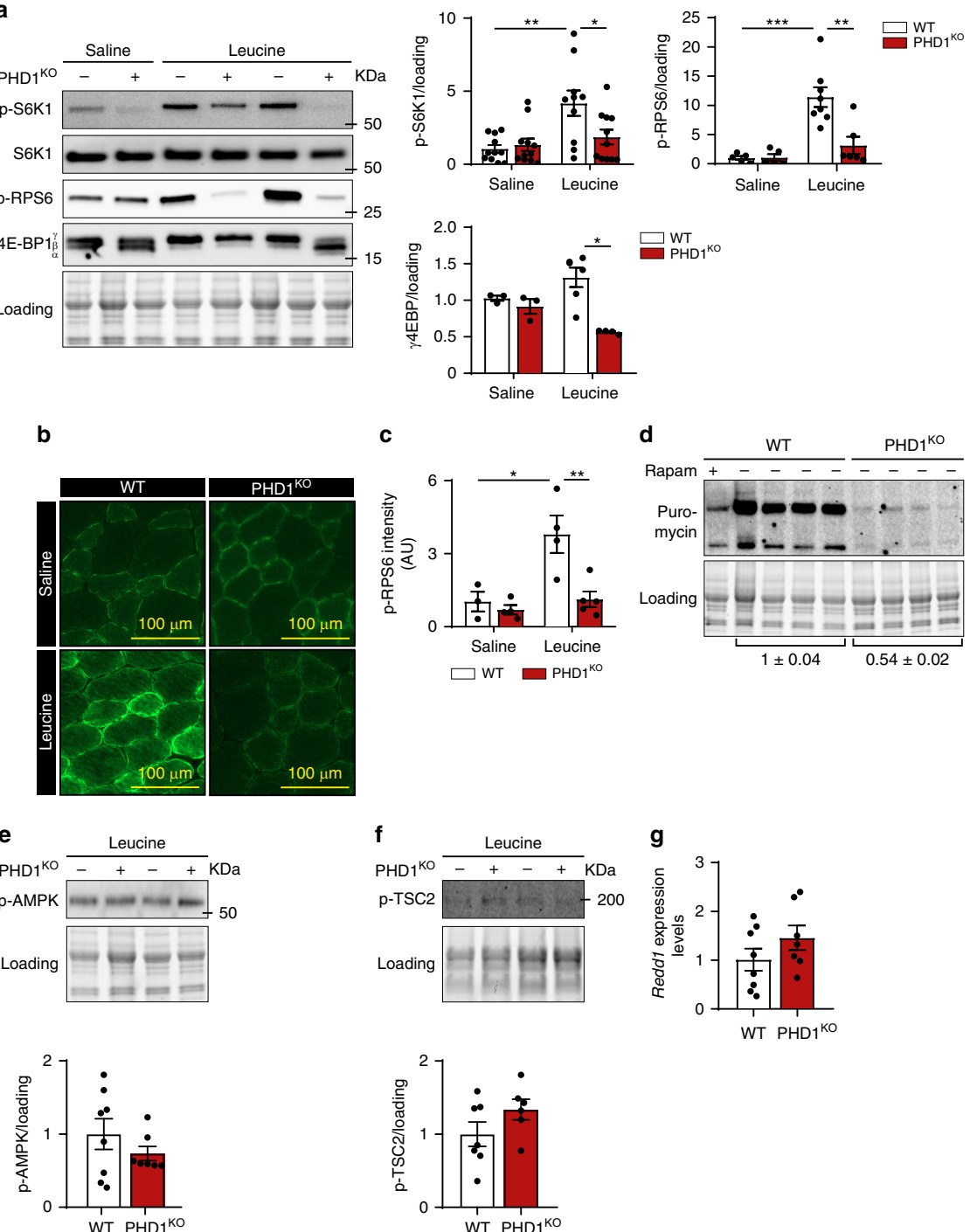

**Fig. 2 PHD1 is required for leucine-mediated mTORC1 activation in vivo. a** Representative pictures (left panel) and quantification (right panels) of western blot analysis of S6K1, RPS6, and 4E-BP1 phosphorylation in TA muscles from WT (white bars) and PHD1KO (red bars) female mice 30 min after saline or leucine gavage. Representative pictures **b** and quantification **c** of p-RPS6 immunofluorescence analysis in TA muscle of WT (white bars) and PHD1KO (red bars) mice 30 min after saline (saline) or leucine (leucine) gavage. Intensity measurements are provided in arbitrary units (AU). **d** Representative pictures and quantification of western blot analysis of puromycin incorporation in TA muscle from WT ($n = 4$) and PHD1KO male mice ($n = 4$) 30 min after leucine gavage. Rapamycin (rapam) was used as a negative control. Representative pictures (top panel) and quantification (bottom panel) of western blot analysis of AMPK phosphorylation (**e**) and TSC2 phosphorylation (**f**) in TA muscle from WT (white bars) and PHD1KO (red bars) mice 30 min after leucine gavage. **g** *Redd1* mRNA expression levels in TA muscle from WT (white bars) and PHD1KO (red bars) female mice. Statistics: two-way ANOVA test with a Holm–Sidak post hoc test (**a**, **c**) or unpaired *t* test (**e–g**) (*$p < 0.05$; **$p < 0.01$; ***$p < 0.001$; ns not significant). Each dot represents a single mouse. Bar graphs represent mean ± SEM (error bars). Data are represented as fold change to WT saline (**a**, **c**, **e**, **f**) or fold change to WT (**g**). TA *m. tibialis anterior*. See also Supplementary Fig. 2. Source data are provided as a Source Data file.

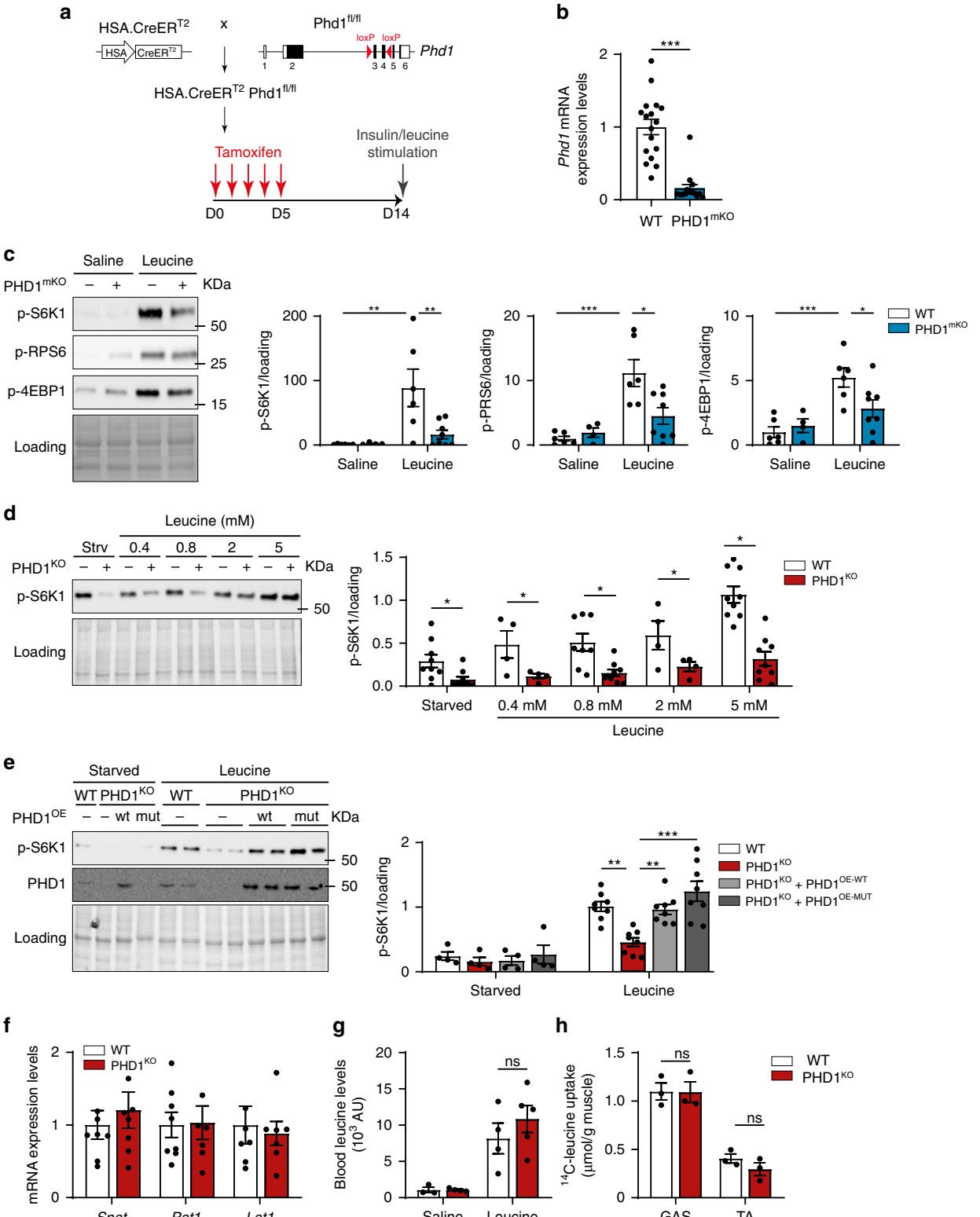

**PHD1 controls intracellular leucine sensing through LRS.**
Upon entering the cell, leucine is "sensed" by SESNs and LRS which transmit intracellular leucine availability towards mTORC1[17,18,22]. To investigate whether PHD1 controls leucine mediated mTORC1 activation via affecting LRS or SESN1–2, we assessed protein levels of LRS and SESN1–2 in muscle. Interestingly, LRS protein levels were lower both in PHD1[KO] as well as PHD1[mKO] muscle when compared to WT animals (Fig. 4a and

Supplementary Fig. 4a). In contrast, SESN1 nor SESN2 protein levels were affected by loss of *Phd1* (Fig. 4b–d and Supplementary Fig. 4b–d).

To further investigate the role of LRS and SESNs, we evaluated whether *Sesn* knockdown or LRS overexpression could rescue leucine-dependent mTORC1 activation in *Phd1*-deficient myotubes. Silencing *Sesn1* nor *Sesn2* changed the responsiveness to leucine PHD1[KO] myotubes (Supplementary Fig. 4e–h).

**Fig. 3 PHD1 controls muscle mass in a cell-autonomous fashion. a** Schematic representation showing the generation of muscle specific *Phd1* deficient mice (PHD1[mKO]) and the experimental protocol. **b** *Phd1* mRNA expression in TA muscle from WT (white bars) and PHD1[mKO] (blue bars) male and female mice. **c** Representative pictures (left panel) and quantification (right panels) of western blot analysis of p-S6K1, p-RPS6, and p-4E-BP1 in TA muscles from WT (white bars) and PHD1[mKO] (blue bars) male and female mice 30 min after saline or leucine gavage. **d** Representative pictures (left panel) and quantification (right panel) of western blot analysis of p-S6K1 in differentiated myotubes from WT (white bars) and PHD1[KO] (red bars) mice after 1 h starvation (strv or starved) or stimulated with increasing concentrations of leucine for 30 min (leucine). **e** Representative pictures (left panel) and quantification (right panel) of western blot analysis of p-S6K1 and PHD1 expression in differentiated WT (with bars), PHD1[KO] (red bars), PHD1[KO] + PHD1[OE-WT] (light gray bars) and PHD1[KO] + PHD1[OE-MUT] (dark gray bars) myotubes after 1 h starvation (starved) or 1 h starvation followed by 30 min stimulation with 5 mM leucine (leucine). **f** mRNA expression of different leucine transporters in TA of WT (white bars; $n = 8$) and PHD1[KO] (red bars $n = 7$) female mice. **g** Blood leucine in WT (white bars) and PHD1[KO] (red bars) mice 30 min after saline or leucine gavage. **h** Muscle leucine uptake in GAS and TA from WT (white bars) and PHD1[KO] (red bars) female mice 30 min after leucine gavage. Statistics: two-way ANOVA with a Holm-Sidak post-hoc test (**c**, **e**) or unpaired *t* test (**b**, **d**, **f**, **g**, **h**) (*$p < 0.05$; **$p < 0.01$; ***$p < 0.001$; ns not significant). Each dot represents a single mouse (**b**, **c**, **g**, **h**) or means of independent experiments (**d**, **e**). Bar graphs represent mean ± SEM (error bars). Data is represented as fold change to WT saline (**c**), to WT leucine 5 mM (**d**, **e**) or to WT (**b**, **f**). See also Supplementary Fig. 3. Source data are provided as a Source Data file.

Conversely, lentiviral overexpression of LRS in PHD1[KO] myotubes restored LRS protein content and increased mTORC1 activation upon leucine stimulation (Fig. 4e). This data indicates that LRS, but not SESN1–2, is involved in PHD1-mediated mTORC1 activation in muscle cells. We therefore decided to focus on LRS and study its role in leucine-dependent mTORC1 activation in muscle.

LRS belongs to a family of proteins known as aminoacyl tRNA synthetases whose canonical function is to ensure that the genetic code is accurately deciphered by attaching the correct amino acid to the equivalent tRNA[20]. However, LRS also serves as a leucine sensor for mTORC1 by functioning as a GTP activating protein for RagD[52], thereby promoting lysosomal translocation of mTORC1. GTP-bound RagD subsequently promotes the LRS-dependent leucylation of RagA at $K^{142}$ ($K_{Leu}$ 142) upon leucine stimulation[21]. Whether LRS levels are controlled by metabolism is, however, not clear. In fact, the leucine sensing ability of LRS in muscle is not described. To confirm that LRS is required for leucine mediated mTORC1 activation in myotubes, we inhibited the interaction of LRS with RagD by using BC-LI-0186[52] and confirmed reduced leucine mediated activation of mTORC1 (Supplementary Fig. 4i). This data shows that LRS is involved in leucine-dependent mTORC1 activation in muscle cells.

To confirm that PHD1 reduces LRS downstream signaling to mTORC1, we stimulated WT and PHD1[KO] myotubes with leucine and evaluated RagA $K_{Leu}$142 levels. In *Phd1*-deficient myotubes, leucine failed to increase RagA $K_{Leu}$142 (Fig. 4e). Moreover, we observed reduced RagA $K_{Leu}$142 levels in vivo (Fig. 4f). We also monitored the lysosomal localization of mTOR by performing immunofluorescent stainings for mTOR and the lysosomal marker LAMP2 under starved and stimulated conditions. Whereas we saw a clear increase in LAMP2/mTOR colocalization in WT myotubes upon (amino acid) stimulation, increased colocalization was not observed in PHD1[KO] myotubes (Fig. 4g, h). The data shows that loss of *Phd1* reduces LRS protein content, leading to impaired mTORC1 translocation to the lysosomes and impaired RagA $K_{Leu}$142.

**PHD1 interacts with LRS and controls LRS stability.** Although PHDs are mainly known for their hydroxylation dependent functions, it has been shown that they interact with other proteins to modulate their stability and/or activity via hydroxylation-independent mechanisms[34]. To study whether PHD1 and LRS interact, we overexpressed both PHD1-flag and LRS-myc in HEK 293T cells and found that LRS and PHD1 coimmunoprecipitated (Fig. 5a). The ability to interact with LRS was independent of PHD1's enzymatic activity, since PHD1[MUT] also interacted with LRS (Fig. 5a). Furthermore, we performed an in vitro hydroxylation assay using recombinant PHD1 and LRS[53] using HIF1α

as a positive control, but did not detect any hydroxylated prolines on LRS, whereas HIF1α was clearly hydroxylated under the same experimental conditions (Fig. 5b and Supplementary Fig. 5a).

Interestingly though, whereas the interaction between PHD1 and LRS was weak under normal culture conditions, it was further promoted when the enzymatic activity of PHD1 was inhibited upon treatment with the hypoxia mimetic, dimethyl 2-oxoglutarate (DMOG), or during amino acid starvation (Fig. 5c). Based on these observations, we wondered whether the interaction between PHD1 and LRS improves LRS stability and whether LRS stability is enhanced when the interaction is promoted, such as during nutrient and oxygen deprivation. We first performed a cycloheximide chase assay for LRS in WT vs. PHD1[KO] myotubes, and noticed that upon translation elongation inhibition, LRS levels dropped faster when *Phd1* is absent, indicating that loss of *Phd1* reduces LRS protein stability (Fig. 5d). Second, we found that LRS protein levels remain remarkably stable (or even increase) during short-term amino acid starvation as well as hypoxia in WT myotubes, but rapidly go down in PHD1[KO] myotubes (Fig. 5e, f and Supplementary Fig. 5b, c) showing that PHD1 is required for the maintenance of LRS stability during oxygen and amino acid deprivation. Altogether, we show that upon metabolic stress, including amino acid starvation and hypoxia when PHD1 enzymatic activity is inhibited, PHD1 interacts with LRS and protects it from degradation in a manner which does not require its enzymatic activity. The protection of LRS during low oxygen and amino acid levels ensures a rapid and efficient activation of mTORC1 as soon as nutrient levels are restored.

**PHD1 levels and LRS activity decline during aging.** Aging is associated with a loss of muscle mass. Moreover, the ability of older muscle to efficiently activate protein synthesis in response to amino acids, a condition termed anabolic resistance, is impaired[54–56]. Several molecular mechanisms underlying anabolic resistance have been proposed, including reduced amino acid delivery and uptake into the muscle[57], but it is not known whether leucine sensing mechanisms are affected. Thus, to explore the relevance of our findings in a human setting, we compared PHD1 protein content in muscle samples from a small cohort where samples were obtained from healthy old ($n = 8$, 4 women and 4 men, 72.6 ± 2.3 y (mean ± SEM)) vs. young ($n = 8$, 4 women and 4 men, 26.1 ± 1.1 y) volunteers after an overnight fast[55]. In this cohort, old people showed reduced activation of muscle protein synthesis in response to milk protein (containing a high dose of leucine) ingestion when compared to young subjects[55]. We found that older muscle on average showed an almost 50% decrease in PHD1 levels (Fig. 6a). Lower PHD1 content was confirmed using immunofluorescent stainings (Fig. 6b, c).

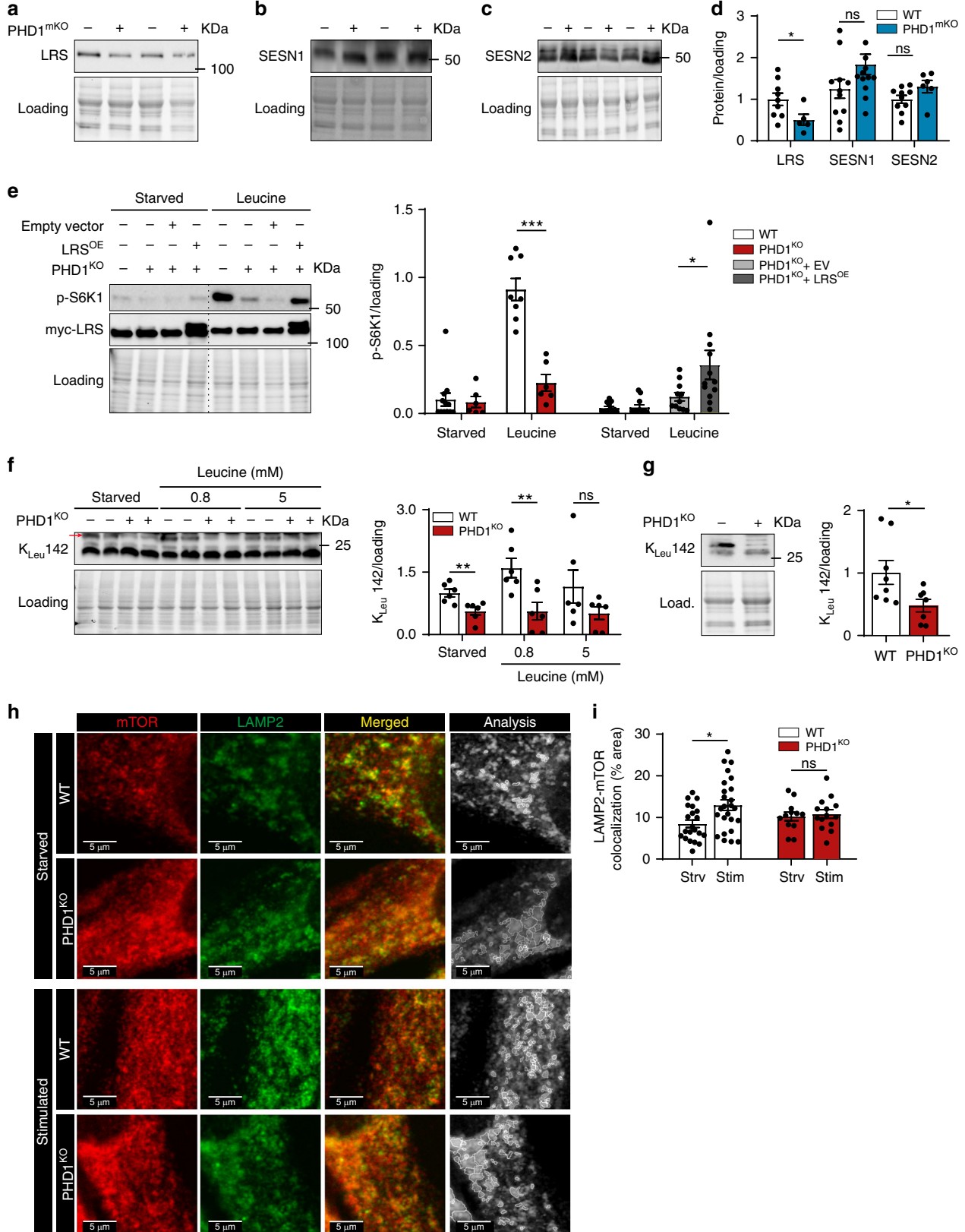

Importantly, in other cell types PHD1 localizes mainly in the nucleus[58,59], but several groups have also reported cytosolic localization[60,61] and function of PHD1[62]. We saw clear cytosolic localization of PHD1 in muscle fibers whereas nuclear staining was detected, but to a much lower extent. LRS levels were also lower but this failed to reach statistical significance because variability between subjects was larger (Fig. 6a). Notwithstanding, we found a clear reduction in LRS induced leucylation of RagA (RagA $K_{Leu}142$) (Fig. 6d), and RagA $K_{Leu}142$ levels correlated with PHD1 levels ($r = 0.56$; $p = 0.02$). Thus, in aged humans, impaired activation of muscle protein synthesis in response to leucine coincides with reduced PHD1 content and LRS activity.

**Fig. 4 PHD1 controls intracellular leucine sensing through leucyl tRNA synthetase. a–c** Representative pictures and quantification (**d**) of western blot analysis of LRS, SESN1, and SESN2 protein levels in TA muscles from WT (white bars) and PHD1[mKO] male and female (blue bars) mice. **e** PHD1[KO] myogenic progenitors were transduced with lentiviruses to over-express myc-LRS (LRS[OE]), an empty vector (EV) was used as control. Representative pictures (left panel) and quantification (right panel) of western blot analysis of S6K1 phosphorylation and myc-LRS expression levels in differentiated WT (white bars), PHD1[KO] (red bars), PHD1[KO] + EV (light gray bars) and PHD1[KO] + LRS[OE] myotubes (dark gray bars) after 1 h starvation (starved) or 30 min stimulation with 5 mM leucine (leucine). **f** Representative picture (left panel) and quantification (right panel) of western blot analysis of RagA leucylation (K_{Leu}142) levels in WT (white bars) and PHD1[KO] (red bars) myotubes after 1 h starvation (starved) or 30 min stimulation with different doses of leucine (leucine). **g** Representative picture (left panel) and quantification (right panel) of western blot analysis of RagA leucylation (K_{Leu}142) levels in TA from WT (white bars) and PHD1[KO] (red bars) female mice. Representative pictures (**h**) and quantification (**i**) of colocalization between mTOR (red) and LAMP2 (green) in WT (white bars) and PHD1[KO] (red bars) myotubes. Merged (gray) picture shows the outline of the colocalization analysis between mTOR and LAMP2 that was performed to generate data shown in panel (**i**). Dots represent quantification of individual myotubes from three independent experiments. Statistics: two-way ANOVA test with a Holm–Sidak post hoc test (**e, i**) or unpaired $t$ test (**d, f, g**) (*$p < 0.05$; **$p < 0.01$; ***$p < 0.001$; ns not significant). Each dot represents a single mouse. Dots represent experimental duplicates from four independent experiments (**e, f**). Bar graphs represent mean ± SEM (error bars). Data is presented as fold change to WT starved (**e, f**) or fold change to WT (**d, g**). See also Supplementary Fig. 4. Source data are provided as a Source Data file.

## Discussion

Protein synthesis is a highly energy consuming process which in many cell types is inhibited during low oxygen and nutrient availability[28,63,64]. On the other hand, mechanisms that allow (and even promote) protein synthesis during hypoxia have also been described as stabilization of HIF2α promotes mTORC1 activation during low amino acid availability by increasing the expression of the amino acid carrier *Lat1*[29]. In fact, in vivo data in lung and liver during hypoxia have indicated that HIF2α mediated activation of mTORC1 can prevail over HIF1α dependent mTORC1 inhibition[29]. It is therefore likely that control of mTORC1 oxygen sensitive mechanisms is dependent on the cellular context. Here, we describe that the oxygen sensor PHD1 promotes mTORC1 activation upon leucine stimulation. We found that PHD1 interacts with the leucine sensor LRS and controls LRS stability. This interaction is promoted under conditions of hypoxia and amino acid starvation when PHD1 hydroxylation activity is inhibited and protects LRS from degradation. Consequently, loss of *Phd1* reduces the stability of LRS, impairs leucine-mediated activation of mTORC1 and leads to lower muscle mass in vivo.

We found that PHD1[KO] mice have lower muscle mass, muscle fiber size, and force production. This was not caused by increased muscle proteolysis as we did not find upregulation of atrogenes nor activation of autophagy. Instead, we observed a blunted mTORC1 activation in response to leucine stimulation leading to lower protein synthesis. mTORC1 plays a central role in the regulation of muscle mass: activation of mTORC1 induces muscle hypertrophy[65,66], whereas skeletal muscle specific inactivation results in low muscle mass coinciding with dystrophy[38]. Furthermore, the fact that sustained activation of mTORC1 in muscle also induces myopathy characterized by muscle atrophy and weakness[67], indicates that mTORC1 activity should be tightly controlled in order to prevent muscle dysfunction. We did not find any evidence for muscle dystrophy or myopathy, showing that the ability of PHD1 to control mTORC1 activity is modest, and is restricted to modulating upstream anabolic events that activate mTORC1. Several anabolic signals such as leucine, growth factors, but also eccentric contractions are potent regulators of mTORC1 in muscle[38,40,68]. The activation of mTORC1 in response to growth factors (such as insulin) as well as contractions was however preserved in PHD1[KO] mice. Instead, there was a selective "resistance" of PHD1[KO] muscle toward leucine, showing that PHD1 controls mTORC1 via events upstream of mTORC1 which are specifically linked to the sensing and transmission of amino acid signals.

Ample literature has shown the contribution of LRS and SESNs to leucine sensing in vitro[17,18]. However, their functional role as leucine sensors in vivo in tissues such as skeletal muscle, has been questioned since the Kd dissociation constant for leucine of both SESN2 and LRS is approximately tenfold lower than the leucine concentration observed in skeletal muscle of humans[22,69,70]. This implies that both enzymes are completely saturated at physiological leucine concentrations and has led to the suggestion that alterations in protein levels of leucine sensors impose an additional level of control in modulating mTORC1 activity[22]. On the other hand, in vitro experiments in HEK293T cells demonstrated that leucylation of RagA, considered as a readout of LRS activity, increases in an almost linear manner with leucine doses up to 30-fold of the normal physiological values[21]. Thus, the exact mechanisms through which leucine sensors control mTORC1 still remain to be fully elucidated. We show here that altering LRS levels suffices to increase leucine mediated mTORC1 activation in myotubes. Moreover, we found decreased LRS levels and lower leucylation of RagA (RagA K_{Leu}142) in PHD1[KO] animals, showing that LRS protein levels and LRS activity can be modulated in vivo to/and control mTORC1 activation.

It has been reported that LRS levels remain stable during amino acid starvation[71]. This is in agreement with our data. In addition, we show that PHD1 is required to maintain LRS protein stability during conditions of oxygen and amino acid deprivation. The increased stability of LRS is in sharp contrast to most amino acid tRNA synthetases, which are rapidly degraded during amino acid starvation induced autophagy[72]. These observations suggest that the maintenance of LRS protein levels may not be related to its role as a tRNA synthetase but with other noncanonical LRS functions. Instead, keeping LRS levels high during episodes of low oxygen and/or amino acid availability may allow the cell to ensure a fast and efficient reactivation of mTORC1 as soon as oxygen and nutrient levels are restored. Our data also shows active modulation of LRS levels by metabolism.

We did not observe differences in SESN1–2 protein content upon loss of *Phd1*, nor did we observe a rescue of leucine mediated mTORC1 activation upon knockdown of SESN1–2 in PHD1[KO] cells. It is important to mention that these data do not exclude a role for SESNs in regulating leucine mediated mTORC1 activation in muscle. *SESN2* knockout mice also show preserved mTORC1 activation in response to insulin, but the response to leucine stimulation has not been evaluated[73]. Moreover, while recent evidence indicates that SESN1, rather than SESN2, is abundantly expressed in muscle and dissociates from GATOR2 in response to oral leucine administration[69], its functional role in mediating mTORC1 activity upon leucine administration is not known. Whether or not SESNs directly interact with other members of the oxygen sensing machinery remains an outstanding question.

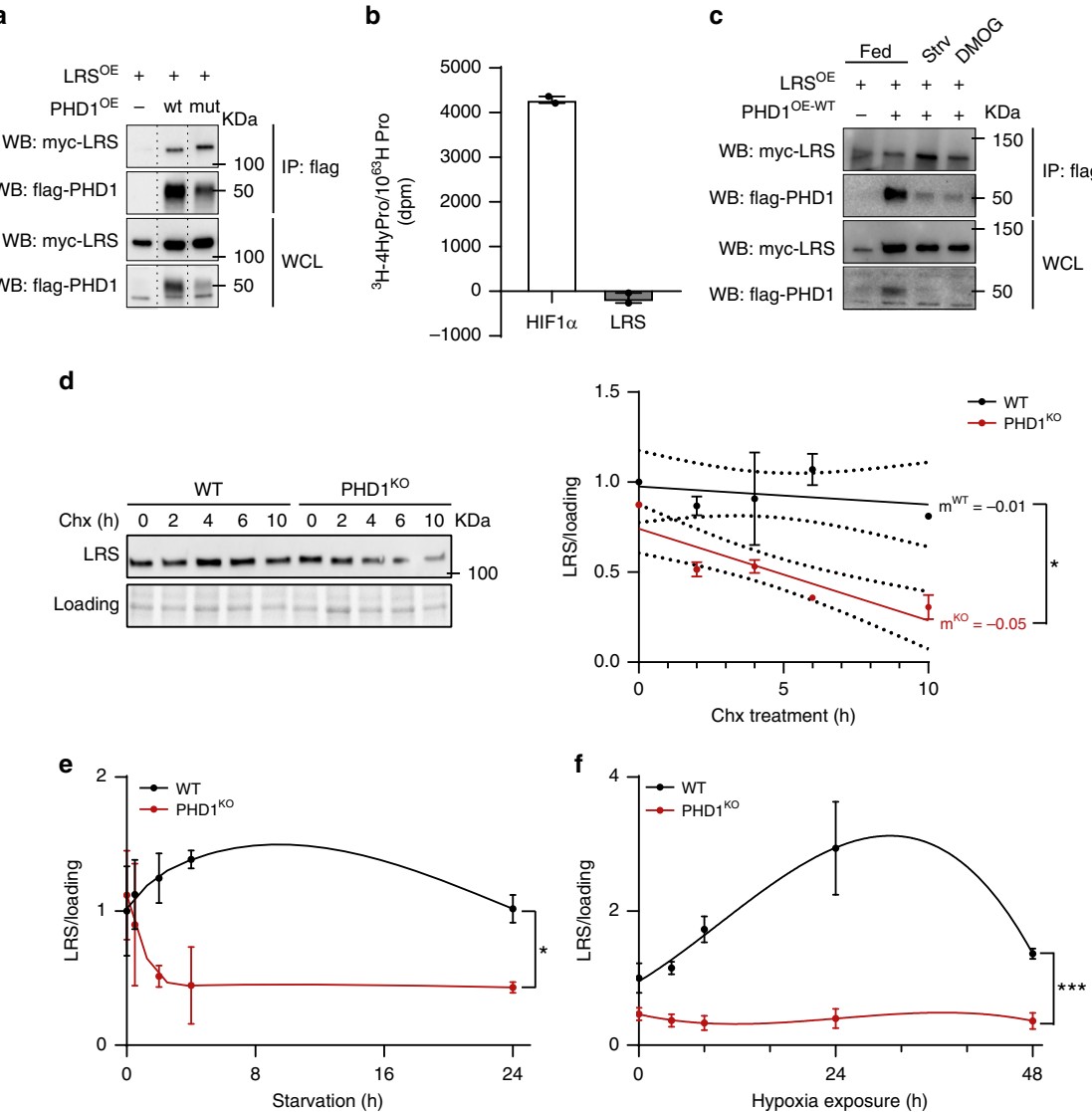

**Fig. 5 PHD1 interacts with LRS and controls LRS stability. a** HEK293T cells were transfected with myc-LRS, myc-LRS and flag-PHD1[WT] or myc-LRS and flag-PHD1[MUT]. Cell lysates were immunoprecipitated with anti-flag antibody. Co-precipitation of myc-LRS was determined by western blot analysis using an anti-myc antibody. The figure shows a representative experiment. **b** In vitro hydroxylation assay using [3]H-labeled in vitro translated LRS (gray bar) or HIF1α (white bar; positive control) in the presence of affinity purified PHD1. **c** HEK293T cells were transfected with myc-LRS or myc-LRS and flag-PHD1[WT] and treated with DMOG or amino acid starved for 4 h. Cell lysates were immunoprecipitated with anti-flag antibody. Co-precipitation of myc-LRS was determined by western blot analysis using an anti-myc antibody. The figure shows a representative experiment. **d** Representative picture (left panel) and quantification (right panel) of western blot analysis of LRS protein levels in WT (black line) and PHD1[KO] (red line) myotubes exposed to cycloheximide (Chx) for different amounts of time. Dots represent the mean from 2 independent experiments. **e** Time-course analysis of LRS protein levels in WT (black line) and PHD1[KO] (red line) myotubes exposed to amino acid starvation. Dots represent the mean from three independent experiments. **f** Time-course analysis of LRS protein levels in WT (black line) and PHD1[KO] (red line) myotubes exposed to hypoxia (1% oxygen). Dots represent the mean from 3 independent experiments. Statistics: two-way ANOVA (in **e** and **f**, interaction effects are indicated) with a a $t$ test (**d**) (*$p < 0.05$; **$p < 0.01$; ***$p < 0.001$; ns not significant). Bar graphs and line graphs represent mean ± SEM (error bars). Data are presented as fold change to WT 0 h (**d**, **e**, **f**). See also Supplementary Fig. 5. Source data are provided as a Source Data file.

PHDs are key proteins mediating oxygen sensing in cells[31,32]. Besides oxygen, PHDs also use αKG, ascorbic acid and Fe$^{2+}$ as cofactors to hydroxylate their targets at specific proline residues[32]. The enzymatic activity of PHDs is lost when one of these co-factors is insufficiently available, such as during hypoxia and/or nutrient deprivation[74,75]. While HIFs are the best characterized targets of PHD hydroxylation activity, more PHD (hydroxylation dependent or independent) targets have been reported[34,76]. In fact, photosynthetic organisms, which produce but do not oxidize oxygen for energy production, have been reported to lack HIFs but not PHDs[77] demonstrating potential HIF-independent functions of PHDs[74].

Our data indicates that PHD1 controls leucine mediated mTORC1 activation in a HIF-independent fashion. First, we did not see transcriptional activation of *Redd1* in PHD1[KO] muscle, a previously described HIF1α-target gene which inhibits mTORC1[15,33,63], nor did we find differences in phosphorylation of its downstream target TSC2. Second, most described hypoxia dependent mechanisms which inhibit mTORC1 have been shown to act via TSC2 (or in the case of AMPK, via RAPTOR

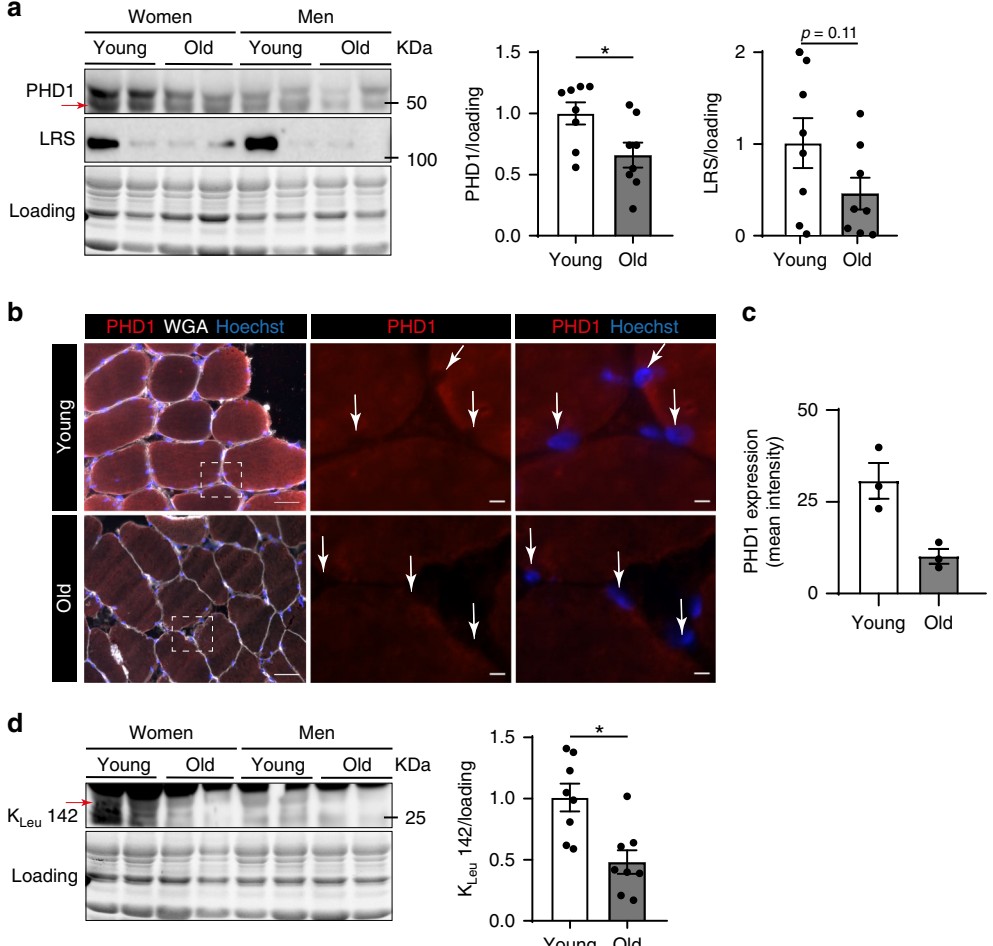

**Fig. 6 PHD1 levels and LRS activity decline during aging. a** Representative picture (left panel) and quantification (right panel) of western blot analysis of PHD1 and LRS protein levels in *m. vastus lateralis* biopsies from young (white bars) and old (gray bars) volunteers. Representative immunofluorescent pictures (**b**) and quantification (**c**) of PHD1 (red), Hoechst nuclear staining (blue) and wheat germ agglutinin (WGA, white) in young and old skeletal muscle. Arrows indicate nuclei. **d** Representative picture (left panel) and quantification (right panel) of western blot analysis of RagA leucylation ($K_{Leu}142$) levels in *m. vastus lateralis* biopsies from young (white bars) and old (gray bars) volunteers. Statistics: unpaired *t* test (**a**) (*$p < 0.05$; **$p < 0.01$; ***$p < 0.001$; ns not significant). Dots represent values from different volunteers. Bar graphs represent mean ± SEM (error bars). Data is presented as fold change to young (**a**, **d**). Source data are provided as a Source Data file.

directly)[30,33,45]. Preserved activation of mTORC1 by insulin (acting via TSC2) as well as contractions (acting via TSC2 or RAPTOR directly) strongly argues against involvement of TSC2 or RAPTOR itself. And last, in agreement with our observations, in vitro reports have shown that PHD dependent control of mTORC1 upon leucine stimulation can occur independent of HIFs/TSC2[74,78]. Indeed, DMOG treatment of amino acid starved MEFs did not result into accumulation of HIF1α or an increase in HIF transcriptional activity. In addition, DMOG treatment also prevented mTORC1 activation in both TSC2 knockout MEFs, suggesting that PHDs act through the Rag GTPases.

The ability of PHD1 to ensure mTORC1 activity upon leucine stimulation was independent of its hydroxylation activity, since reintroduction of the catalytic dead PHD1 in PHD1^KO myotubes completely restored mTORC1 activity. Second, we found that PHD1 interacted with LRS and this interaction also did not require PHD1 hydroxylation activity because the enzymatic dead mutant PHD1 also interacted with LRS. Due to different levels of overexpression for wild type vs. mutant PHD1 within our immunoprecipitation experiments, it is difficult to make a clear statement on whether inhibition of the enzymatic activity of PHD1 suffices to actively promote the interaction or whether

other unknown processes are involved. Nonetheless, we found that reducing PHD activity by using DMOG as well as amino acid starvation increased the interaction with LRS and preserved LRS stability in WT cells. Consequently, loss of *Phd1* prevented LRS stabilization under conditions of amino acid and oxygen shortage, and resulted into a rapid decline of LRS protein levels. The detailed mechanisms underlying this improved interaction require further investigation. Indeed, our data does not allow us to conclude that the interaction between LRS and PHD1 is actively promoted when PHD1 hydroxylation activity is inhibited. An alternative explanation could be that reduced interaction of PHD1 with its canonical target HIF during hypoxia increases the availability of PHD1 to interact with other substrates. Nonetheless, similar observations were made in breast cancer cells were the interaction between PHD1 and NRF1/PGC1α to preserve mitochondrial function during tumor growth was enhanced during hypoxia and was also independent of PHD1's hydroxylation activity[79]. Our findings show that PHDs can exert different functions during normoxia/nutrient availability when they hydroxylate HIFs (and potentially other targets) vs. hypoxia/nutrient depletion when they execute an autonomous role within the hypoxia-dependent program.

Altogether, PHD1 integrates oxygen and nutrient availability to modulate the activation of mTORC1 in response to leucine. The in vivo relevance of our data is underscored by the lower muscle mass of PHD1[KO] animals. Moreover, we observed that PHD1 levels and LRS activity are lower in muscle of elderly with anabolic resistance. To the best of our knowledge, we did not find any report showing reduced PHD1 levels in aging muscle. The upstream mediators of altered PHD1 levels during aging are not known and warrant further research. But since we show that modulating PHD1/LRS levels can alter the muscular response to leucine, our data raises the question whether PHD1/LRS levels could be therapeutically targeted to improve the anabolic effect of leucine and to prevent the development of age-related sarcopenia.

## Methods

**Reagents and cell culture**. HEK293T cells (DSMZ, ACC 635) were maintained in DMEM (ThermoFisher Scientific; 41966052) supplemented with 10% fetal bovine serum (FBS) (ThermoFisher Scientific, 10270-106). Freshly isolated myogenic progenitors (MPs) were cultured in a 1:1 ratio of DMEM (ThermoFisher Scientific, 12320032) and Ham's F-10 (1×) nutrient mix (ThermoFisher Scientific, 22390058) supplemented with 10% horse serum (ThermoFisher Scientific, 16050-122), 20% FBS and 10 ng/ml basic-FGF (ThermoFisher Scientific, PHG0266). All media were supplemented with 100 units/ml penicillin and 100 μg/ml streptomycin. Cells were routinely cultured in 21% $O_2$ and 5% $CO_2$ (normoxic conditions). For hypoxic exposure, differentiated myotubes were placed into a hypoxia workstation at 1% $O_2$. All cells were regularly tested for absence of mycoplasma. Lentiviral LRS-overexpression vector was purchased from Origene (ref number: RC221682L3). Retroviral vectors to overexpress PHD1 and PHD1-H358A were purchased form Addgene (catalog numbers: 22704 and 22705, respectively). Lentiviral vectors to knockdown SESN1 and SESN2 were purchased form Sigma (catalog numbers SESN1: TRCN0000087808 and SESN2: TRCN0000087791). Cyclohexamide, puromycin, DMOG, and polybrene were from Sigma (c7698, P8833, D3695, and TR1003).

Isolation and culture of primary MPs: Muscle tissue was digested in HBSS supplemented with 1.5% bovine serum albumin (BSA) and 2 mg/mL collagenase type II (ThermoFisher Scientific, 17101015) for 1 h at 37 °C. After centrifugation, the cell pellet was then filtered using 40 and 100 μm cell strainers and a heterogeneous cell population was purified by FACS sorting or by serial preplating. For FACS, MPs were sorted based on positive alpha 7-integrin (1:100) and absence of Sca1, CD31, and CD45 (1:1000, see Supplementary Table 1). For serial preplating, a freshly extracted heterogeneous cell population was preplated for 1.5 h at day 0, 4, and 10 after cell extraction. MPs were cultured in growth medium (see above) on dishes coated with Matrigel Basement Membrane Matrix (Corning, #356237, 1/25 dilution). When cells reached 80% confluency, growth medium was switched to differentiation medium containing low-glucose DMEM, 2% HS and 1% P/S. MPs were fully differentiated after 3 days.

Leucine stimulation: differentiated myotubes were starved for 1 h in low-glucose DMEM lacking all amino acids (US Biology, D9800-13), supplemented with 10% dialyzed FBS (dFBS) (ThermoFisher Scientific, 26400-044). Leucine was supplemented for 30 min in the indicated concentrations. For the RagA leucylation experiments, differentiated myotubes were incubated with DMEM containing all AA, except leucine (Sigma, 9443) + 10% FBS for 16 h supplemented with 0, 0.8, or 5 mM leucine.

Insulin stimulation: differentiated myotubes were serum starved for 16 h and 100 nM insulin was added to the medium for 5, 10, 20, and 30 min.

Amino acid starvation: differentiated myotubes were starved from all amino acids in low-glucose amino acids free DMEM (US Biology, D9800-13) supplemented with 10% dFBS for 0.5, 2, 4, and 24 h.

Cyclohexamide (CHXChx) time-course: differentiated myotubes were treated with 100 μg/ml Chx (Sigma-aldrich, c7698) for 2, 4, 6, 8, and 10 h.

**Viral infections**. For lentiviral infections HEK293T cells were transfected with 8 μg of each lentiviral vector and 5 μg of pmd2.G and pCMV8.9 lentivirus packaging plasmids using Lipofectamine 2000 (Invitrogen, 11668019). For retroviral infections, HEK293T cells were transfected with 10 μg of each retroviral vector and 10 μg of pCL-ECO retroviral packaging plasmids. Cell culture supernatants were harvested 24, 48, and 72 h after transfection, passed through a 0.45 μm pore size filter, supplemented with 4 μg/μl of polybrene and added to MPs.

**In vitro pull-down assay**. HEK293T were transfected with Myc-LRS[18] and flag-PHD1 for 72 h and were subsequently lysed (50 mM Tris-HCl, pH 7.4, 10 mM NaCl, 1 mM EDTA, 0.5 mM EGTA, 1 mM MgCl2, 0.1% CHAPS and 0.5% Triton X-100 supplemented with protease inhibitors). Cell lysates were centrifuged at 15,000g for 20 min. A total of 300 μg protein from the supernatant fraction was incubated with ANTI-FLAG® M2 Affinity Gel (Sigma, A2220-1ML) at 4 °C

overnight. After washing 3 times in lysis buffer supplemented with 50 mM NaCl, the precipitates were dissolved in laemmli buffer.

**In vitro hydroxylation assay**. The potential ability of PHD1 to hydroxylate LRS was studied by determining the amount of radioactive 4-hydroxyproline formed in the substrate. Full-length cDNAs for human LRS and HIF1α (positive control) were translated in the presence of L-[2,3,4,5-3 H]-proline (85 mCi/mmol, Perkin Elmer) in a rabbit reticulocyte lysate using a TNT® Quick Coupled Transcription/ Translation System (Promega). The products were analyzed by 10% sodium dodecyl sulfate polyacrylamide gel electrophoresis followed by fluorography. Totally, 45 μl of the products, containing approximately $2 \times 106$ cpm of incorporated radioactive proline, was used as the substrate for the affinity purified PHD1[80] in a final reaction volume of 0.25 ml under reported conditions[80] except that the 2-oxoglutarate was non-labeled. PHD1 was omitted from the control reactions. The samples were subsequently dialyzed to remove any remaining free L-[2,3,4,5-3 H]-proline and the radioactive 4-hydroxyproline formed in the substrate was analyzed in hydroxylyzed samples by means of a specific radiochemical assay[53].

**Human muscle samples**. Participants were informed of the purpose and methodology of the study prior to providing written informed consent. Ethical approval was obtained through the NHS Black country Research Ethics Committee (13/WM/ 0429)[55]. The study was performed in agreement with the standards set by the Declaration of Helsinki (seventh edition). Muscle samples from old ($n = 8$, 4 women and 4 men, 72.6 ± 2.3 y) and young ($n = 8$, 4 women and 4 men, 26.1 ± 1.1 y) volunteers were obtained from m. vastus lateralis after an overnight fast. Details from biopsy procedure and freezing method can found elsewhere[55].

**Animals**. Whole body Phd1 knockout (PHD1[KO]) mice (50% Swiss/50% 129S1 background) were previously generated[35]. Phd1[fl/fl] mice were generated using homologous recombination in embryonic stem (ES) cells. The targeting vector was built in a pPNTlox2 plasmid with loxP sites flanking the neomycine resistance (neor) cassette, and contained from 5′ to 3′: a 5.3-kb EcoRV-HindIII fragment comprising exon 2 and the majority of intron 2 as 5′ homology arm, a 1.8-kb floxed neor cassette, a 2.5-kb HindIII-EcoRV fragment comprising exon 3 through 6, in which a third loxP site was introduced downstream of exon 4 along with a novel EcoRI site for genotyping purposes, an SV40 thymidine kinase expression cassette for negative selection purposes. The introduced third loxP site together with the loxP site located 3′ of the neor cassette thus flanked the Phd1 gene segment comprising exon 3 and 4, which encode part of the catalytic domain conferring the prolyl hydroxylase activity. The construct was linearized with NotI and electroporated into G4 ES cells of 129SvEv/C57BL6 origin (kind gift from A. Nagy, Toronto)[35]. After positive–negative drug selection with 200 μg/ml G418 (Invitrogen) and 2 μM Ganciclovir (Sigma Aldrich), resistant clones were analyzed for correct homologous recombination by appropriate Southern blotting and PCR. Correctly targeted ES cells were then transiently electroporated with the pOG231 Cre-recombinase expressing plasmid to excise the floxed neor cassette. Clones with exclusive excision of the neor cassette and retention of the floxed genomic fragment, were identified by PCR and Southern blot screens based on the retention of the newly introduced EcoRI site. The resulting Phd1[+/fl] ES cells were then used for diploid aggregation with Swiss morula embryos to obtain chimeric mice, and germline Phd1[+/fl] offspring from crosses of male chimeras with C57BL6/J females were backcrossed to C57BL6/J mice for 6 generations, to generate Phd1[+/fl] mice with a genetic background of >98% C57BL6/J. To generate muscle-specific Phd1 knockout (PHD1mKO) animals, Phd1[fl/fl] mice (C57BL/6J background) were crossed with transgenic mice expressing Cre under the control of HSA promotor[46] and kept in a homozygous mating system for fl/fl and heterozygous for HSA.Cre-negative littermates were used as controls. At the age of 10 weeks, mice were injected with 1 mg tamoxifen for 5 days and a washout period of 9 days was allowed before experiments were initiated. To generate HSA.iCre-Rosa[mTmG] mice, transgenic mice expressing Cre under the HSA promotor were crossed with heterozygous Rosa[mTmG] mice[47].

**Experimental procedures**. All animal procedures were approved by the Veterinary office of the Canton of Zürich (licence nr. ZH255-16), by the local ethics committee of the KU Leuven, Belgium (P174-2014) and were executed in compliance with the institutional and national guidelines and regulations. Sample size was determined based on previous experiments in our lab and similar studies reported in the literature. All mice used for the experiments were housed 3–4 littermates per cage in individually ventilated cages at standard housing conditions (22 °C, 12 h light/dark cycle, dark phase starting at 7 pm), with ad libitum access to chow (18% proteins, 4.5% fibers, 4.5% fat, 6.3% ashes, Provimi Kliba SA) and water. Health status of all mouse lines was regularly monitored according to FELASA guidelines.

Leucine and insulin administration in vivo: The morning (8:00 A.M.) of the experiments, mice were fasted for 4 h. A suspension of 40 g L-leucine/L in distilled water was freshly prepared and mice were administered 0.4 g L-leucine kg$^{-1}$ bw$^{-1}$ via oral gavage. Control mice were administered equal volume saline (0.9% NaCl). Exactly 30 min after saline or leucine administration, mice were sacrificed and GAS, SOL, TA, and EDL, were harvested and either snap frozen in liquid $N_2$

(biochemical analysis) or embedded in Tissue-Tek and frozen in liquid $N_2$ cooled isopentane (histological analysis). Blood samples were obtained from the tail vain, centrifuged and serum was stored at −80 °C for further analysis. For insulin stimulation experiments, 0.2 IU/kg insulin was intraperitoneally injected and animals were sacrificed for muscle dissection exactly 15 min after.

*Muscle force measurements:* WT and PHD1[KO] mice were sacrificed and *soleus* was excised and placed in Krebs–Henseleit buffer (120 mM NaCl, 4.8 mM KCl, 25 mM NaHCO3, 2.5 mM CaCl₂, 1.2 mM KH2PO4, 2 mM MgSO4) supplemented with 1× MEM amino acid mixture (Invitrogen, 11130051) and 25 mM glucose. After 15 min of equilibration, *soleus* was attached to the lever arms of an Aurora system (Aurora Scientific) and submerged in Krebs–Henseleit buffer that was continuously gassed with 95% $O_2$/5% $CO_2$ and maintained at 37 °C. Muscle length was adjusted until a single stimulus pulse elicited maximum force during a twitch (Lo) under isometric conditions. After a 5 min rest, a force frequency protocol was initiated by subsequently providing a pulse train (lasting 250 ms) of 1–30–50–80–150–250 and 300 Hz with 1 min rest between every intensity. Data were analyzed using the software provided by the manufacturer.

*MRI:* body composition was measured in 12–14-week-old PHD1[KO] mice after short term food withdrawal (4 h) using EchoMRI-100H (EchoMRI) according to the manufacturer's instructions.

**Immunohistochemistry and histology.** *Fiber typing:* skeletal muscle sections (10 μm) were dried and washed for 5 min in phosphate-buffered saline (PBS) supplemented with 0.05% triton (PBST) and subsequently blocked for 60 min in PBST + 10% goat serum (ThermoFisher Scientific, 16200-064). Afterwards a primary antibody cocktail was applied for 120 min for myosin heavy chain I (1:50), IIa (1:200) and b (1:100) (see Supplementary Table 1) diluted in PBST + 10% goat serum. After washing 3 times for 5 min, a secondary antibody cocktail, diluted in PBST + 10% goat serum, was applied for goat anti-mouse Alexa Fluor 488,350,568 and wheat germ agglutinin Alexa fluor 647 (1:250) (ThermoFisher Scientific, see Supplementary Table 2) for 60 min. Slides were mounted after a 3 × 5 min wash, sealed with glass cover slips and imaged with a epifluorescent microscope (Zeiss Axio observer Z.1) at 10×. Fiber cross-sectional area was automatically determined with a Muscle J plugin for Image J software[81]. Muscle fiber type was manually counted on tile scans. To determine the presence of centrally nucleated fibers, sections were stained with haematoxylin and eosin.

mTORC1 downstream signaling in mouse skeletal muscle and PHD1 in human skeletal muscle: 9 μm thin TA sections were fixed −20 °C acetone for 10 min. After incubating in PBS for 15 min, the sections were then incubated in solution A (PBS with 5% goat serum and 0.3% CHAPS) for 1 h. After three 5 min washes with PBS, samples were incubated overnight at 4 °C with PBS with 0.5% BSA and 0.3% CHAPS containing primary antibodies specific for pS6[ser235/236] (1:500) (see Supplementary Table 1). The next morning, after three 5 min washes in PBS, samples were incubated for 1 h in solution A with the appropriate 488-conjugated goat anti-rabbit IgG secondary antibody (1:250, ThermoFisher Scientific, see Supplementary Table 2). Images were captured with Zeiss Axio observer Z.1. Exposure times were kept constant for each image and color intensity was set to the WT saline condition.

LAMP2-mTORC1 colocalization: WT and PHD1[KO] myoblasts were differentiated in matrigel-coated (1:25) 8-well chambers (ibidi; 80826). Briefly, fully differentiated myotubes were untreated, starved for 1 h in HBSS or starved in HBSS for 1 h and restimulated with differentiation medium for 1 h, after which they were rinsed with ice-cold (4 °C) PBS once and fixed for 5 min with 4% paraformaldehyde in PBS. The chamber wells were rinsed twice with PBS and permealized with 0.2% Triton X-100 in PBS for 15 min. After rinsing twice with PBS, the wells were blocked for one hour in blocking buffer (0.25% BSA in PBS) and incubated with primary antibody for LAMP2 and mTOR (1:200) in blocking buffer overnight at 4 °C (see Supplementary Table 1). Subsequently, the wells were rinsed twice with blocking buffer and incubated with secondary antibodies (diluted in blocking buffer 1:500) for one hour at room temperature in dark. Individual myotubes were imaged with 60× confocal microscope (Olympus olympus fluoview FV 3000). LAMP2-mTOR colocalization was determined by ImageJ using a plugin described by Moser et al.[82]. This analysis combines object based-recognition with pixel-intensity-correlation. The combined colocalization area (in pixels) of threshold 1 and 2 was multiplied by Pearson R and divided by the total fiber area.

**SuNSET analysis for protein synthesis.** WT and PHD1[KO] mice were gavaged with 0.4 g L-leucine kg−1 BW−1 and simultaneously IP-injected with 0.040 μmol/g puromycin (Sigma-Aldrich, P8833) dissolved in 100 μl of PBS. Mice were sacrificed 30 min after injection, muscles were snap-frozen in liquid $N_2$ and stored for further analysis.

**Western blot.** Muscle tissue (10–15 mg) was homogenized with a tissue homogenizer (Omni THq) in ice cold lysis buffer (1:15 w/v): [50 mM Tris–HCl pH 7.0, 270 mM sucrose, 5 mM EGTA, 1 mM EDTA, 1 mM sodium orthovanadate, 50 mM glycerophosphate, 5 mM sodium pyrophosphate, 50 mM sodium fluoride, 1 mM DTT, 0.1% Triton-X 100 and a complete protease inhibitor tablet (Roche Applied Science)]. The same lysis buffer was used for differentiated myotubes. Homogenates were centrifuged at 10,000g for 10 min at 4 °C. Supernatant was collected

and protein concentration was measured using the DC protein assay kit. Totally, 20–40 μg of total protein was loaded in a 15-well pre-casted gradient gel (Bio-rad, 456-8086). After electrophoresis, a picture of the gel was taken under UV-light to determine protein loading using strain-free technology. Proteins were transferred via semidry transfer onto a polyvinylidene fluoride membrane (Bio-rad, 170-4156) and subsequently blocked for 1 h at room temperature with 5% milk in TBS-Tween. Membranes were incubated overnight at 4 °C with primary antibodies listed in Supplementary Table 1 (1:500–1:1000). The appropriate secondary antibodies (1:5000) for anti-rabbit and anti-mouse IgG HRP-linked antibodies (Cell signaling, see Supplementary Table 2) were used for chemiluminescent detection of proteins. Membranes were scanned with a chemidoc imaging system (Bio-rad) and quantified using Image lab software (Bio-rad).

**RT-qPCR.** Muscle tissue (10–15 mg) was homogenized with a tissue homogenizer (Omni THq) in 1000 μl ice-cold TRIzol (ThermoFisher Scientific, 15596018), and after addition of 200 μl chloroform, the homogenate was spun down for 15 min at 12,000g. The clear phase was mixed with 70% Ethanol and transferred into a mRNA extraction column (ThermoFisher Scientific, 12183018A). Subsequently, mRNA was extracted according to the manufacturer's instruction. Messenger RNA of cells was extracted using the same mRNA extraction kit. The purity and quantity of mRNA was assed via a photospectometer (Tecan, Spark). mRNA was reverse-transcribed using iSctipt cDNA synthesis kit (Bio-Rad, 170-8891). A SYBER Green-based master mix (ThermoFisher Scientific, A25778) was applied for real-time qPCR analysis. Primers that were used are listed in Supplementary Table 3. To compensate for variations in mRNA input and efficiency of reverse-transcription, three housekeeping genes were used (*GAPDH, β-Actin*). The delta–delta $C_T$ method was used to normalize the data.

**GC-MS determination of leucine levels.** Extractions for subsequent mass-spectroscopy analysis were prepared and analyzed as previously described[83].

**Skeletal muscle leucine uptake.** After a 6 h fasting period, mice were gavaged with a bolus of 0.16 g/kg L-Leucine spiked with 1.5 μCi L-[14 C(U)]-Leucine (PerkinElmer, NEC279E050UC). Blood samples were obtained from the tail vein before and 3, 6, 10, 15, 20, and 30 min after administration of the solution. Once the final blood sample was taken, mice were sacrificed and GAS and TA were quickly dissected and processed for further analysis. Muscles were digested in 0.5 ml 1 M NaOH for 30 min at 65 °C, vortexed to assure complete digestion and neutralized with 0.5 ml 1 M HCl. Radioactivity was determined in 400 μl of this mixture in duplicate or in 5 μl serum by liquid scintillation counting (Beckman LS 6500; Beckman Coulter). The rate of L-leucine uptake (Kin) was calculated by the equation Kin = total dpm muscle/AUC 0–30 min.

**Statistical analysis.** A two-way ANOVA design was used to assess the statistical significance of differences between mean values over phenotype and treatment. When appropriate, Tukey post hoc test was used. To determine statistical significances between two groups, an unpaired student's *t* test was used. Level of significance was set at $\alpha = 0.05$. Results are shown as mean ± SEM.

**Reporting summary.** Further information on research design is available in the Nature Research Reporting Summary linked to this article.

## Data availability
The data presented in this study are available from the corresponding authors upon reasonable request. The source data underlying Figs. 1–6 and Supplementary Figs. 1–5 are provided as a Source Data file.

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

## Acknowledgements

We gratefully acknowledge Paola Gilardoni, Andrew Palmer, and Axel Moens for the technical assistance in the lab.

## Author contributions

G.D. and I.S.-A. designed the study, conducted the experiments, performed the laboratory analysis, wrote and edited the paper. E.M., K.V., P.K. and G.F. conducted the experiments and performed the laboratory analysis. B.S., and L.B. provided the human muscle samples. P.K. performed the in vitro hydroxylation experiments. S.K., L.D., B.B., P.C. and S.-M.Z. provided the reagents/materials and experimental protocols. K.D.B., conceptualized and designed the study and wrote the paper.

## Competing interest

The authors declare no competing interests.
