## [Peer Review File · Nature Communications]

Reviewers' comments:

Reviewer #1 (Remarks to the Author):

In this study, D'Ulst and colleagues report that PHD1 controls muscular mass by regulating the stability of the leucine sensor LRS. This novel PHD1 function is independent of the enzymatic properties of PHD1 as a proline hydroxylase. The authors use an elegant combination of in vivo and in vitro approaches to test their working hypothesis.

The study is novel, highly significant, and with translational implications. The quality of the data is excellent. The authors' conclusions are supported by the data as shown.

No significant weaknesses could be identified.

Minor point:

Figure 6: It would be helpful if the authors could perform immunohistochemistry for PHD1 on human muscular tissue isolated from either young or old donors to provide information about the spatial localization of the signal.

--

Reviewer #2 (Remarks to the Author):

D'Hulst et. al, sought to determine the contribution of HIF prolyl hydroxylase isoform 1 (PHD1) in regulating skeletal mass. The authors utilized a germline deleted as well as an inducible skeletal muscle-specific PHD1 knockout mice models. The authors observed a slight decrease in skeletal muscle mass which contributes to a decrease in lean mass and a shift towards smaller muscle cross sectional area. The authors also report a blunted response to protein synthesis activation with leucine administration in the PHD1 KO. The authors next report that the function of PHD1 was not dependent on its hydroxylase activity. The authors next show that the leucyl tRNA synthetase (LRS) leucine sensor and not Sestrin2 was decreased in the muscle-specific PHD1 KO animals. The authors show that PHD1 interacts with LRS and helps to stabilize it. In addition they report that under conditions of starvation or hypoxia in the absence of PHD1, LRS levels are lower. Lastly, the author report that PHD1, LRS and leucylation levels were decreased in older human muscle biopsies. Taken together the authors concluded that PHD1 integrates oxygen and nutrient availability to activate mTORC1 in response to leucine.

Author Comments

A well written manuscript, highlighting the contribution of PHD1 in maintaining muscle mass. The authors have highlighted a new link between PHD1, mTORC activation and leucyl tRNA synthetase in skeletal muscle. Overall a well-executed study.

Specific points

Figure 1. The authors should provide some discussion as to why the reduction in lean mass only contributed to the decrease in body mass in males only. To this point the authors should include the percent fat mass for the PHD KO males and female animals. Lastly, the authors should specify in the figure legends whether males or females are being used for the in vivo assessment for the purpose of clarity.

Figure 3. Did the authors observe any changes in muscle mass in the inducible muscle-specific PHD1 knockout animals? While the effect on S6K phosphorylation and other mechanistic components appears to be conserved in the mKOs, it would be good to understand whether the reduction in muscle mass also occurs.

--

Reviewer #3 (Remarks to the Author):

In this manuscript, D'Hulst et al evaluate the consequences of PHD1 deficiency on skeletal muscle mass and performance. Through the use of several novel mouse models, the investigators highlight a non-canonical role for PHD1 (independent of its hydroxylase function) in the regulation of mTORC1 activation by leucine. The authors perform both in vivo and in vitro experiments that clearly show that loss of PHD1 prevents leucine activation of mTORC1. The cell autonomous effects are confirmed in an inducible skeletal muscle specific knockout of PHD1. Mechanistically, the authors suggest leucyl tRNA synthetase as a potential link between PHD1 and mTORC1. Although interesting, there are several limitations and concerns that should be addressed that would further support the main conclusions of this study.

Major concerns:

-Most of the focus is on LRS acting as a leucine sensor for mTORC1. This is based on the observation that in PHD1KO muscle there is a reduction in LRS protein content, accelerated degradation. Moreover, overexpression of LRS in culture increases mTORC1 activity. In addition, the authors note an unremarkable change in the other "putative" leucine sensor, Sestrin2. However, Sestrin2 expression is lowest in muscle and recent reports indicate Sestrin1 acts as a leucine sensor for mTORC1 in muscle (PMID: 30835510). The authors only look at Sestrin 2 (Figure 4B) which nevertheless appears to have increased Sestrin2 content in PHD1KO samples. A more detailed analysis of all Sestrins and their potential functional role as a link between PHD1 and mTORC1 must be explored further.

-Figure 5G, there is a substantial disconnect between the western blot and quantitation regarding the role of LRS OE and mTORC1 activity. For instance, starved levels of pS6K1 in LRS OE/PHD1KO appear much enhanced compared to wild-type and PHD1KO EV yet the bar graph indicates no increase?

-In the skeletal muscle specific knockout, muscle weights and leucine-stimulated protein synthesis (via puromycin) should be measured and reported to confirm cell autonomous effect

-Figure 3C, given the mild effect on pS6K compared to whole body knockout, more detailed analysis of mTORC1 signaling (4EBP1, pS6) should be performed

-The study in human tissue as is does not add anything. Rather, it suggest that a reduction in PHD1 is not correlated with a decrease in LRS in human tissue. Moreover, recent reports suggest mTORC1 activity is elevated in aged muscle in rats (Joseph, Giselle et al BioRxiv 2019). Do the authors see a similar effect on mTORC1 activity in their samples?

Minor concerns:

-Figure 4F, the ICC staining is not very convincing and should be improved

-PHD1 protein levels in knockout should be reported

Detailed reply to the reviewers

Reviewer #1:

In this study, D'Ulst and colleagues report that PHD1 controls muscular mass by regulating the stability of the leucine sensor LRS. This novel PHD1 function is independent of the enzymatic properties of PHD1 as a proline hydroxylase. The authors use an elegant combination of in vivo and in vitro approaches to test their working hypothesis. The study is novel, highly significant, and with translational implications. The quality of the data is excellent. The authors' conclusions are supported by the data as shown.

No significant weaknesses could be identified.

We would like to thank the reviewer for this positive evaluation of our work.

Minor point:

Figure 6: It would be helpful if the authors could perform immunohistochemistry for PHD1 on human muscular tissue isolated from either young or old donors to provide information about the spatial localization of the signal.

To address the reviewer's comment, we have performed immunofluorescent stainings to get further insight into the spatial localization of PHD1. To do so, we used cryosections from young and old human skeletal muscle. In the original manuscript, we showed by western blot that PHD1 protein levels are lower in muscle samples from old subjects (see Figure 6). To validate our data, we have performed immunofluorescent staining for PHD1 in muscle cryosections from the same subjects. We could confirm the lower overall intensity of PHD1 in old muscles using signal intensity quantifications of stained muscles (see figure 6B). Also, PHD1 has been shown to be localized mainly in the cell nucleus (Metzen et al., 2003; Ortmann et al., 2016; Steinhoff et al., 2009) but several groups have also reported cytosolic localization (Bur et al., 2018; Couvelard et al., 2008; Moser et al., 2013; Soilleux et al., 2005; Zhang et al., 2015) and cytosolic functions (Moser et al., 2013) of PHD1. As highlighted in Figure 6B, we found clear cytosolic localization of PHD1 in muscle fibers of adult human subjects. Nuclear staining was detected, but to a lower extent. Immunofluorescent staining for PHD1 in cultured myoblasts confirmed both nuclear and cytosolic localization (See figure RL 1). Of note, we could confirm the specificity of our antibody to detect PHD1 by the observation that there is a weaker signal in PHD1^{KO} myotubes compared to the signal in WT myotubes/myoblasts (See figure RL1). Altogether, our data shows that in human skeletal muscle and mouse primary myoblasts/myotubes, PHD1 is located in the cytosol as well as (to a lower extent) the nucleus.

We have included information about spatial localization of PHD1 in the text (p14, line 304-308):

'Lower PHD1 content was confirmed using immunofluorescent stainings (Figure 6B, and 6C). Importantly, in other cell types PHD1 has been shown to be localized mainly in the nucleus⁸⁸⁻⁹⁰, but several groups have also reported cytosolic localization⁹¹⁻⁹⁵ and function of PHD1⁹³. We saw clear cytosolic localization of PHD1 in muscle fibers whereas nuclear staining was detected, but to a much lower extent.'

Figure RL 1. Immunofluorescent stainings on PHD1 and Hoechst in WT and PHD1^{KO} myoblasts. Scale bare represents 10 μ m.

Reviewer #2:

D'Hulst et. al, sought to determine the contribution of HIF prolyl hydroxylase isoform 1 (PHD1) in regulating skeletal mass. The authors utilized a germline deleted as well as an inducible skeletal muscle-specific PHD1 knockout mice models. The authors observed a slight decrease in skeletal muscle mass which contributes to a decrease in lean mass and a shift towards smaller muscle cross sectional area. The authors also report a blunted response to protein synthesis activation with leucine administration in the PHD1 KO. The authors next report that the function of PHD1 was not dependent on its hydroxylase activity. The authors next show that the leucyl tRNA synthetase (LRS) leucine sensor and not Sestrin2 was decreased in the muscle-specific PHD1 KO animals. The authors show that PHD1 interacts with LRS and helps to stabilize it. In addition they report that under conditions of starvation or hypoxia in the absence of PHD1, LRS levels are lower. Lastly, the author report that PHD1, LRS and leucylation levels were decreased in older human muscle biopsies. Taken together the authors concluded that PHD1 integrates oxygen and nutrient availability to activate mTORC1 in response to leucine.

A well written manuscript, highlighting the contribution of PHD1 in maintaining muscle mass. The authors have highlighted a new link between PHD1, mTORC activation and leucyl tRNA synthetase in skeletal muscle. Overall a well-executed study.

We would like to thank the reviewer for the positive evaluation of our work.

Specific points

Figure 1. The authors should provide some discussion as to why the reduction in lean mass only contributed to the decrease in body mass in males only. To this point the authors should include the percent fat mass for the PHD KO males and female animals. Lastly, the authors should specify in the figure legends whether males or females are being used for the in vivo assessment for the purpose of clarity.

We agree with the reviewer's comment and have now included the percentage fat mass (both males and females) in the manuscript (Figure 1C and S1C) and accordingly adapted the results section in the manuscript (p6, line 94-98). As reported before (Thomas et al., 2016), genetic loss of *Phd1* leads to an increase in both subcutaneous as well as visceral fat mass. In our mice, the increase in FM was higher in female mice (more than doubling of percent fat mass) when compared to male mice (50% increase in percent fat mass). Unfortunately, only male mice were included in the aforementioned study by Thomas et al. In our female mice, the higher increase in fat mass compensated for the lower muscle weights in the PHD1^{KO} animals, overall resulting in the absence of a significant reduction in body weight. It is difficult to speculate about the underlying cause of these gender specific differences in adipose tissue mass upon loss of *Phd1*. It is known that female mice have different body temperature, different heart rate, different oxygen consumption rate, different adipose tissue biology and can respond differently to metabolic stress (Chang et al., 2018; Landsberg et al., 2009; Yang et al., 2007).

Following the reviewer's last comment, we have now adapted all figure legends and included a statement about whether males/females (or mixed) were used. We would like to mention that we observed reduced leucine mediated activation of mTORC1 in both males as well as females in both PHD1^{KO} and PHD1^{mKO} mice. While the data included in the original manuscript was obtained in females only, we now also provide experimental data showing blunted increase in mTORC1 activation in PHD1^{KO} muscles upon leucine administration in males as well. Therefore, we can conclude that the ability of PHD1 to control leucine mediated mTORC1 activity is independent of gender. We have included those data in the supplements (figure S2, panel A) and added a statement in the text (line 134-136):

'Inhibition of mTORC1 activation upon deletion of Phd1 in response to leucine stimulation was observed in both female (Figure 2A) as well as male mice (Figure S2A), so both genders were used for subsequent experiments.'

Figure 3. Did the authors observe any changes in muscle mass in the inducible muscle-specific PHD1 knockout animals? While the effect on S6K phosphorylation and other mechanistic components appears to be conserved in the mKOs, it would be good to understand whether the reduction in muscle mass also occurs.

We thank the reviewer for this comment. To address the reviewer's comment we have included below (Figure RL2) the weight for TA and GAS muscle from WT and PHD1^{mKO} mice. We did not observe any decrease in muscle weight (TA or GAS) at this time point in the inducible muscle-specific PHD1 knockout mice (see Figure RL2). However, we would like to mention that, since (i) gene recombination was only induced less than two weeks before the experiments were performed and (ii) loss of *Phd1* only reduces the leucine-dependent increase in mTORC1 and not contraction/insulin response, we did not expect changes in muscle mass at this time point. In this line, a recent report using muscle-specific *Raptor* deficient mice (where mTOR –downstream signaling is severely blunted in response to any stimulus) showed no decrease in muscle mass 21 days after recombination (You et al., 2018).

Figure RL 2. Muscle weight of male (A) and female (B) WT and PHD1^{mKO} mice 10 days after tamoxifen injections

Nonetheless, to address whether inducible muscle deletion of PHD1 affected muscle mass, we tried to mimic the full KO condition by inducing muscle deletion at early postnatal stages. To do so, we injected HSA-Cre-ER^{T2}^{-/-} x *Phd1*^{fl/fl} (WT) and HSA-Cre-ER^{T2}^{+/-} x *Phd1*^{fl/fl} mice (PHD1^{mKO}) females with tamoxifen (5 x 0.1 mg per mouse at P13-14). Since it is known that there is still significant contribution of muscle stem cells to muscle growth during and maintenance up till 14-16 postnatal weeks (Bruusgaard et al., 2006; Murach et al., 2017; Pallafacchina et al., 2013; Pawlikowski et al., 2015; White et al., 2010), we applied additional Tamoxifen injections every two weeks (1 x 1 mg per mouse, see scheme in figure RL3). Subsequently, mice were sacrificed at identical age when compared to the PHD1^{KO} experiments (16 weeks). We found that TA and GAS weight were lower in PHD1^{mKO} compared to WT (11.3% and 14.3% respectively), but due to lower variation, only TA reached significance (p = 0.03 for TA and p = 0.2 in GAS - See figure RL3). Even though these data underscore the role of PHD1 in muscle growth, we prefer not to include them in the manuscript. The main reason for this is that the HSA-Cre-ER^{T2}^{+/-} inducible mouse model is not commonly used to follow up the effect of gene deletions during growth. Moreover, the repeated use of Tamoxifen might interfere with hormonal household of the mice, which prevents us from making sound and convinced statements about our observations.

Figure RL 3. Muscle weight of WT and PHD1^{mKO} mice after repetitive tamoxifen injections starting at age 13 days. (A) Experimental setup. (B) Muscle weights of WT and PHD1^{mKO} mice at age 16 weeks.

Reviewer #3:

In this manuscript, D'Hulst et al evaluate the consequences of PHD1 deficiency on skeletal muscle mass and performance. Through the use of several novel mouse models, the investigators highlight a non-canonical role for PHD1 (independent of its hydroxylase function) in the regulation of mTORC1 activation by leucine. The authors perform both in vivo and in vitro experiments that clearly show that loss of PHD1 prevents leucine activation of mTORC1. The cell autonomous effects are confirmed in an inducible skeletal muscle specific knockout of PHD1. Mechanistically, the authors suggest leucyl tRNA synthetase as a potential link between PHD1 and mTORC1. Although interesting, there are several limitations and concerns that should be addressed that would further support the main conclusions of this study.

Major concerns:

-Most of the focus is on LRS acting as a leucine sensor for mTORC1. This is based on the observation that in PHD1KO muscle there is a reduction in LRS protein content, accelerated degradation. Moreover, overexpression of LRS in culture increases mTORC1 activity. In addition, the authors note an unremarkable change in the other "putative" leucine sensor, Sestrin2. However, Sestrin2 expression is lowest in muscle and recent reports indicate Sestrin1 acts as a leucine sensor for mTORC1 in muscle (PMID: 30835510). The authors only look at Sestrin 2 (Figure 4B) which nevertheless appears to have increased Sestrin2 content in PHD1KO samples. A more detailed analysis of all Sestrins and their potential functional role as a link between PHD1 and mTORC1 must be explored further.

We thank the reviewer for this valuable concern. As requested by the reviewer, we provide a more in-depth analysis of the other SESTRINs, their role in leucine mediated activation of mTORC1 and their potential interaction with PHD1. First, we measured mRNA as well as protein expression levels of *SESN 1-3* (mRNA) in PHD1^{KO} and PHD1^{mKO} muscle (used for *in vivo* experiments) as well as in myotubes derived from primary myoblasts that were isolated from WT and PHD1^{KO} mice. loss of PHD1 *in vivo* did not affect mRNA nor protein levels of SESN1-3. We do acknowledge that there was a slight increase in the (protein) levels of SESN1 in *m. tibialis anterior* of PHD1^{mKO}, however this failed to reach statistical significance, and we also did not observe any difference in the PHD1^{KO}. In addition, we did not find any difference in SESN1 protein content in primary myotubes, arguing against an active regulation of SESNs expression by loss of *Phd1*. We have included the protein expression levels data of SESN1 and 2 in the main manuscript (Figure 4B-D and S4B-D, line 229-230)

Edited in manuscript (line 229-230): '*In contrast, SESN1 nor SESN2 protein levels were affected by loss of Phd1 (Figure 4B-D and S4B-D).*'

Figure RL 4. SESN1-3 expression in WT and PHD1^{(m)KO} muscle and primary myotubes. (A) SESN1-2-3 mRNA in WT and PHD1^{KO} muscle. (B) SESN1-3 mRNA in WT and PHD1^{mKO} muscle. (C) SESN1-2 protein levels in WT and PHD1^{KO} muscle. (D) SESN1-2 protein levels in WT and PHD1^{mKO} muscle. setup.

Notwithstanding, to further investigate whether SESNs are involved in PHD1 mediated leucine-induced mTORC1 activation, we silenced both SESN1 and 2 in primary myoblasts derived from PHD1^{KO} muscle, leading to a ~70% knock down efficiency (see Figure S4E-F). Subsequently, we generated myotubes from these cells and repeated our leucine stimulation experiments (identical conditions as the ones described in the original manuscript). Interestingly, silencing of SESN1 or SESN2 in PHD1^{KO} myotubes failed to rescue leucine mediated mTORC1 activation (Figure S4G-H).

Overall, since loss of *Phd1* does not affect SESNs protein levels and silencing SENS1 or SESN2 in PHD1^{KO} myotubes doesn't rescue leucine-dependent mTORC1 activation to the WT levels, our data do not support a role for SESNs in the PHD1-mTORC1 signaling pathway. We would however like to highlight (as we also did in the original version of the manuscript) that this data does not indicate that SESNs are not involved in leucine mediated mTORC1 activation in muscle. Off note, we tested several short hairpin sequences and found that knock down of SESN1 induced a significant compensatory upregulation of other SESNs, which underscores the crucial role of these genes in myoblasts/myotubes (See figure S4E-F).

We included the data on SESN1 and 2 knock-down in figure S4 and edited the revised manuscript (Line 231-234):

'To further investigate the role of LRS and SESNs, we evaluated whether SESN knock-down or LRS overexpression could rescue leucine-dependent mTORC1 activation in Phd1 deficient myotubes. Silencing SESN1 nor SESN2 changed the responsiveness to leucine in myotubes derived from PHD1KO muscle (Figure S4E-H).'

And adapted the discussion section (line 387-391):

'We did not observe differences in SESN1-2 protein content upon loss of Phd1, nor did we observe a rescue of leucine mediated mTORC1 activation upon knockdown of these genes in PHD1KO cells. It is however important to mention that these data do not exclude a role for SESNs in regulating leucine mediated mTORC1 activation in muscle.'

-Figure 5G, there is a substantial disconnect between the western blot and quantitation regarding the role of LRS OE and mTORC1 activity. For instance, starved levels of pS6K1 in LRS OE/PHD1KO appear much enhanced compared to wild-type and PHD1KO EV yet the bar graph indicates no increase?

We agree with the reviewer that the representative image does not completely overlap with the quantitation, and apologize for this mistake. We have now added a western blotting image which fully represents the obtained data (See Figure 4E). We would like to emphasize that the quantification, and thus the interpretation of the data which we presented in the original manuscript are/were correct.

Of note, to improve the coherence of the revised manuscript, the data have now been moved to figure 4. In this section, both SENS knock-down experiments as well as LRS overexpression experiments are presented.

-In the skeletal muscle specific knockout, muscle weights and leucine-stimulated protein synthesis (via puromycin) should be measured and reported to confirm cell autonomous effect

We thank the reviewer for this comment. See also our reply to Reviewer#2. To address the reviewer's comment we have included below (Figure RL2) the weight for TA and GAS muscle from WT and PHD1^{mkO} mice. We did not observe any decrease in muscle weight (TA or GAS) at this time point in the inducible muscle-specific PHD1 knockout mice (see Figure RL2). However, we would like to mention that, since (i) gene recombination was only induced less than two weeks before the experiments were performed and (ii) loss of *Phd1* only reduces the leucine-dependent increase in mTORC1 and not contraction/insulin response, we did not expect major changes in muscle mass at this time point. In this line, a recent report using muscle-specific *Raptor* deficient mice (where mTOR –downstream signaling is severely blunted in response to any stimulus) showed no decrease in muscle mass 21 days after recombination (You et al., 2018).

Nonetheless, as requested by the reviewer, we did measure leucine-stimulated protein synthesis (via puromycin incorporation) in PHD1^{mkO} mice, and observed a 56% reduction in leucine induced protein synthesis in the PHD1^{mkO} compared to WT littermate controls. This data confirms the observations we made in the PHD1^{KO} animals and provide evidence for a cell autonomous effect of loss of *Phd1*. We adjusted the supplementary figure (S3C) and added this to the result section (line 184-185).

Edited in the revised manuscript: *'Moreover, leucine-induced protein synthesis assessed via puromycin incorporation was reduced as well (Figure S3C).'*

Nonetheless, to address whether inducible muscle deletion of PHD1 affected muscle mass, we tried to mimic the full KO condition by inducing muscle deletion at early postnatal stages. To do so, we injected

HSA-Cre-ER^{T2}^{-/-} × *Phd1*^{fl/fl} (WT) and HSA-Cre-ER^{T2}^{+/-} × *Phd1*^{fl/fl} mice (PHD1^{mKO}) females with tamoxifen (5 × 0.1 mg per mouse at P13-14). Since it is known that there is still significant contribution of muscle stem cells to muscle growth during and maintenance up till 14-16 postnatal weeks (Bruusgaard et al., 2006; Murach et al., 2017; Pallafacchina et al., 2013; Pawlikowski et al., 2015; White et al., 2010), we applied additional Tamoxifen injections every two weeks (1 × 1 mg per mouse, see scheme in figure RL3). Subsequently, mice were sacrificed at identical age when compared to the PHD1^{KO} experiments (16 weeks). We found that TA and GAS weight were lower in PHD1^{mKO} compared to WT (11.3% and 14.3% respectively), but due to lower variation, only TA reached significance (p = 0.03 for TA and p = 0.2 in GAS - See figure RL3). Even though these data underscore the role of PHD1 in muscle growth, we prefer not to include them in the manuscript. The main reason for this is that the HSA-Cre-ER^{T2}^{+/-} inducible mouse model is not commonly used to follow up the effect of gene deletions during growth. Moreover, the repeated use of Tamoxifen might interfere with hormonal household of the mice, which prevents us from making sound and convinced statements about our observations.

-Figure 3C, given the mild effect on pS6K compared to whole body knockout, more detailed analysis of mTORC1 signaling (4EBP1, pS6) should be performed.

We agree with the reviewer and have performed a more detailed analysis of mTORC1 downstream signaling (including, p-RPS6 and p-4E-BP1). Our new data confirms and extends the data on p-S6K1 which was included in the initial manuscript. Data are now included in figure 3 (panel C) and will replace the data on mTOR ser2448 phosphorylation since there is debate about whether or not this phosphorylation site mirrors downstream mTORC1 signaling (Figueiredo et al., 2017).

-The study in human tissue as is does not add anything. Rather, it suggest that a reduction in PHD1 is not correlated with a decrease in LRS in human tissue. Moreover, recent reports suggest mTORC1 activity is elevated in aged muscle in rats (Joseph, Giselle et al BioRxiv 2019). Do the authors see a similar effect on mTORC1 activity in their samples?

We acknowledge the reviewer's comment, but respectfully disagree that the human data does not add anything to the manuscript. While due to high inter-subject variation and low subject number the reduction in LRS protein content did not reach statistical significance, we did find reduced LRS activity (assessed as RAGa K_{leu} 142 leucylation) in the human samples and this correlated with PHD1 protein content, r = 0.56; p = 0.02, see line 312). Also, we are aware of recent reports showing increased mTORC1 activity or protein synthesis in aged mouse muscle (Joseph et al., 2019; Miller et al., 2019), but this concerns basal (unfortunately, the exact dietary status of the mice was not reported) protein synthesis, and changes in protein synthesis upon leucine stimulation were not investigated. Concerning the reviewer's question about baseline mTORC1 activity, we kindly refer to the original publication describing the human study (Smeuninx et al., 2017): here, the authors did not observe changes in baseline mTORC1 activity. Nevertheless, we repeated the blot for p-S6K1 and p-RPS6 in the muscle lysates we received from Smeuninx et al. and found a very weak, to almost absent signal for both phosphorylated kinases (see RL 5) in young as well as aged subjects. This is not unexpected because subjects were fasted overnight before the muscle sample collection. Thus, at the specific dietary conditions when muscle biopsies were harvested, we did not see an increase in basal mTORC1 signaling in old human muscle. We would like to emphasize that the inclusion of the human dataset did not aim to address mechanisms underlying the regulation of basal mTORC1 signaling. Rather, they show that an impaired activation of muscle protein synthesis in response to leucine (a phenomenon observed in humans and described as anabolic resistance and well evidenced in literature - see for instance the references we included in the manuscript line 294) coincides with reduced PHD1 content

and LRS activity. Even though we fully realize that these data do not imply causality, they are consistent with and extend our observations in mice.

Figure RL 5. Representative western blots of mTORC1 downstream signaling proteins in young and old subjects (+ means positive control).

Minor concerns:

Figure RL 5. Downstream mTORC1 signaling in young and old skeletal muscle lysates after an overnight fast.

-Figure 4F, the ICC staining is not very convincing and should be improved

We thank the reviewer's comment. We have now included an additional merged picture in Figure 4E to make the representative pictures clearer for the reader.

-PHD1 protein levels in knockout should be reported

We thank the reviewer for their comment. First, we already included in the original manuscript that we observed a 90% decrease in *Phd1* mRNA in PHD1^{mKO} muscle when compared to WT (line 180). Given the contribution of other non-muscle cells to muscle samples, we were there confident to state that *Phd1* was indeed knocked out, also in our muscle specific mouse model. This data is strengthened by our observations showing that PHD1^{mKO} muscle have an identical phenotype when compared to PHD1^{KO} muscle: 1) PHD1^{mKO} muscle show impaired leucine mediated mTORC1 activation; 2) our new data showing that PHD1^{mKO} muscle have lower protein synthesis upon leucine stimulation; 3) PHD1^{mKO} muscle retains its responsiveness to insulin; 4) PHD1^{mKO} muscle has lower LRS protein content.

Unfortunately, we have been facing major difficulties in detecting mouse PHD1 protein in tissue samples using western blotting while it easily detects human PHD1 - a problem that is very well known in the field. In fact, we tried to validate several commercially available antibodies on muscle samples from our whole body PHD1^{KO} animals (which cannot have any PHD1 protein since the DNA has been removed), but were not successful in generating high-quality western blotting images which would allow us to make a reliable statement about PHD1 protein levels (see figure RL6). Nevertheless, for

unknown reasons, we were able to pick up PHD1 in primary myotubes from WT and PHD1^{KO} mice: Figure 3E of the manuscript has a representative western blot for PHD1 protein levels, which clearly shows **absent** PHD1 protein levels in the KO cells and a recovery in the PHD1 overexpressing cells. In addition, we performed immunostainings in WT and PHD1^{KO} myoblasts, and could show a low signal in the *Phd1* deficient cells (see our reply to reviewer #1, Figure RL1) compared to WT. Thus, because we are uncertain of the reliability of the PHD1 ab in mouse muscle lysates, we would like to not add data on PHD1 protein content in the main manuscript.

Figure RL 6. Representative western blots of PHD1 in mouse muscle lysates from WT and PHD1^{KO}/PHD1^{mKO}.

- Aragónés, J., Schneider, M., Van Geyte, K., Fraisl, P., Dresselaers, T., Mazzone, M., et al. (2008). Deficiency or inhibition of oxygen sensor Phd1 induces hypoxia tolerance by reprogramming basal metabolism. *Nat. Genet.* 40, 170–180. doi:10.1038/ng.2007.62.
- Bruusgaard, J. C., Liestø, K., and Gundersen, K. (2006). Distribution of myonuclei and microtubules in live muscle fibers of young, middle-aged, and old mice. *J. Appl. Physiol.* 100, 2024–2030. doi:10.1152/jappphysiol.00913.2005.
- Bur, H., Haapasaari, K., Turpeenniemi-hujanen, T., and Kuittinen, O. (2018). Strong Prolyl Hydroxylase Domain 1 Expression Predicts Poor Outcome in Radiotherapy-treated Patients with Classical Hodgkin’s Lymphoma. *Anticancer Res.* 38, 329–336. doi:10.21873/anticancer.12226.
- Chang, E., Varghese, M., and Singer, K. (2018). Gender and Sex Differences in Adipose Tissue. *Curr. Diab. Rep.* 18. doi:10.1007/s11892-018-1031-3.
- Couvelard, A., Deschamps, L., Rebours, V., Sauvanet, A., Gatter, K., Pezzella, F., et al. (2008). Overexpression of the oxygen sensors PHD-1, PHD-2, PHD-3, and FIH is associated with tumor aggressiveness in pancreatic endocrine tumors. *Clin. Cancer Res.* 14, 6634–6639. doi:10.1158/1078-0432.CCR-07-5258.
- Figueiredo, V. C., Markworth, J. F., and Cameron-Smith, D. (2017). Considerations on mTOR regulation at serine 2448: implications for muscle metabolism studies. *Cell. Mol. Life Sci.* 74, 2537–2545. doi:10.1007/s00018-017-2481-5.
- Joseph, G. A., Wang, S., Zhou, W., Kimble, G., Tse, H., Eash, J. K., et al. (2019). Inhibition of mTORC1 signaling in aged rats counteracts the decline in muscle mass and reverses multiple parameters of muscle signaling associated with sarcopenia. *Mol. Cell. Biol.*, 591891. doi:10.1101/591891.
- Landsberg, L., Young, J. B., Leonard, W. R., Linsenmeier, R. A., and Turek, F. W. (2009). Do the obese have lower body temperatures? A new look at a forgotten variable in energy balance. *Trans. Am. Clin. Climatol. Assoc.* 120, 287–295.

- Metzen, E., Berchner-Pfannschmidt, U., Stengel, P., Marxsen, J., Stolze, I., Klinger, M., et al. (2003). Intracellular localisation of human HIF-1 α hydroxylases: implications for oxygen sensing. *J. Cell Sci.* 116, 1319–1326. doi:10.1242/jcs.00318.
- Miller, B. F., Baehr, L. M., Musci, R. V., Reid, J. J., Peelor, F. F., Hamilton, K. L., et al. (2019). Muscle-specific changes in protein synthesis with aging and reloading after disuse atrophy. *J. Cachexia. Sarcopenia Muscle*, jcs.12470. doi:10.1002/jcs.12470.
- Moser, S. C., Bensaddek, D., Ortmann, B., Maure, J. F., Mudie, S., Blow, J. J., et al. (2013). PHD1 Links Cell-Cycle Progression to oxygen sensing through hydroxylation of the centrosomal protein cep192. *Dev. Cell* 26, 381–392. doi:10.1016/j.devcel.2013.06.014.
- Murach, K. A., White, S. H., Wen, Y., Ho, A., Dupont-Versteegden, E. E., McCarthy, J. J., et al. (2017). Differential requirement for satellite cells during overload-induced muscle hypertrophy in growing versus mature mice. *Skelet. Muscle* 7, 1–13. doi:10.1186/s13395-017-0132-z.
- Ortmann, B., Bensaddek, D., Carvalhal, S., Moser, S. C., Mudie, S., Griffis, E. R., et al. (2016). CDK-dependent phosphorylation of PHD1 on serine 130 alters its substrate preference in cells. *J. Cell Sci.* 129, 191–205. doi:10.1242/jcs.179911.
- Pallafacchina, G., Blaauw, B., and Schiaffino, S. (2013). Role of satellite cells in muscle growth and maintenance of muscle mass. *Nutr. Metab. Cardiovasc. Dis.* 23, S12–S18. doi:10.1016/j.numecd.2012.02.002.
- Pawlikowski, B., Pulliam, C., Betta, N. D., Kardon, G., and Olwin, B. B. (2015). Pervasive satellite cell contribution to uninjured adult muscle fibers. *Skelet. Muscle* 5, 1–13. doi:10.1186/s13395-015-0067-1.
- Smeuninx, B., McKendry, J., Wilson, D., Martin, U., and Breen, L. (2017). Age-related anabolic resistance of myofibrillar protein synthesis is exacerbated in obese inactive individuals. *J. Clin. Endocrinol. Metab.* 102, 3535–3545. doi:10.1210/jc.2017-00869.
- Soilleux, E. J., Turley, H., Tian, Y. M., Pugh, C. W., Gatter, K. C., and Harris, A. L. (2005). Use of novel monoclonal antibodies to determine the expression and distribution of the hypoxia regulatory factors PHD-1, PHD-2, PHD-3 and FIH in normal and neoplastic human tissues. *Histopathology* 47, 602–610. doi:10.1111/j.1365-2559.2005.02280.x.
- Steinhoff, A., Pientka, F. K., Möckel, S., Kettelhake, A., Hartmann, E., Köhler, M., et al. (2009). Cellular oxygen sensing: Importins and exportins are mediators of intracellular localisation of prolyl-4-hydroxylases PHD1 and PHD2. *Biochem. Biophys. Res. Commun.* 387, 705–11. doi:10.1016/j.bbrc.2009.07.090.
- Thomas, A., Belaidi, E., Aron-Wisnewsky, J., van der Zon, G. C., Levy, P., Clement, K., et al. (2016). Hypoxia-inducible factor prolyl hydroxylase 1 (PHD1) deficiency promotes hepatic steatosis and liver-specific insulin resistance in mice. *Sci. Rep.* 6, 24618. doi:10.1038/srep24618.
- White, R. B., Biérinx, A., Gnocchi, V. F., and Zammit, P. S. (2010). Dynamics of muscle fibre growth during postnatal mouse development, White, Biérinx, Gnocchi, Zammit.pdf.
- Yang, J. N., Tiselius, C., Daré, E., Johansson, B., Valen, G., and Fredholm, B. B. (2007). Sex differences in mouse heart rate and body temperature and in their regulation by adenosine A1 receptors. *Acta Physiol.* 190, 63–75. doi:10.1111/j.1365-201X.2007.01690.x.
- You, J.-S., McNally, R. M., Jacobs, B. L., Privett, R. E., Gundermann, D. M., Lin, K.-H., et al. (2018). The role of raptor in the mechanical load-induced regulation of mTOR signaling, protein synthesis, and skeletal muscle hypertrophy. *FASEB J.*, fj.201801653RR. doi:10.1096/fj.201801653RR.
- Zhang, J., Wang, C., Chen, X., Takada, M., Fan, C., Zheng, X., et al. (2015). EglN2 associates with the

NRF 1- PGC 1 α complex and controls mitochondrial function in breast cancer . *EMBO J.* 34, 2953–2970. doi:10.15252/emj.201591437.

REVIEWERS' COMMENTS:

Reviewer #1 (Remarks to the Author):

In this revised manuscript, the authors have satisfactorily addressed reviewers' comments and concerns.

Reviewer #2 (Remarks to the Author):

The manuscripts has improved and the authors have addressed all of this reviewer's concerns.

Reviewer #3 (Remarks to the Author):

The authors have addressed all of my concerns and did a very nice job in the resubmission. Congratulations on the important study.

Reply to the reviewers

REVIEWERS' COMMENTS:

Reviewer #1 (Remarks to the Author):

In this revised manuscript, the authors have satisfactorily addressed reviewers' comments and concerns.

Reviewer #2 (Remarks to the Author):

The manuscripts has improved and the authors have addressed all of this reviewer's concerns.

Reviewer #3 (Remarks to the Author):

The authors have addressed all of my concerns and did a very nice job in the resubmission. Congratulations on the important study.

We thank the reviewers for their excellent contribution to the manuscript